# Structural Information-based Hierarchical Diffusion for Offline Reinforcement Learning

**Xianghua Zeng[1], Hao Peng[1], Yicheng Pan[1], Angsheng Li[1,2], Guanlin Wu[3]**

[1] State Key Laboratory of Complex & Critical Software Environment, Beihang University, Beijing, China
[2] Zhongguancun Laboratory, Beijing, China
[3] National University of Defense Technology, Changsha, China
`{zengxianghua, penghao, yichengp, angsheng}@buaa.edu.cn,`
`wuguanlin16@nudt.edu.cn`

## Abstract

Diffusion-based generative methods have shown promising potential for modeling trajectories from offline reinforcement learning (RL) datasets, and hierarchical diffusion has been introduced to mitigate variance accumulation and computational challenges in long-horizon planning tasks. However, existing approaches typically assume a fixed two-layer diffusion hierarchy with a single predefined temporal scale, which limits adaptability to diverse downstream tasks and reduces flexibility in decision making. In this work, we propose **SIHD**, a novel **S**tructural **I**nformation-based **H**ierarchical **D**iffusion framework for effective and stable offline policy learning in long-horizon environments with sparse rewards. Specifically, we analyze structural information embedded in offline trajectories to construct the diffusion hierarchy adaptively, enabling flexible trajectory modeling across multiple temporal scales. Rather than relying on reward predictions from localized sub-trajectories, we quantify the structural information gain of each state community and use it as a conditioning signal within the corresponding diffusion layer. To reduce overreliance on offline datasets, we introduce a structural entropy regularizer that encourages exploration of underrepresented states while avoiding extrapolation errors from distributional shifts. Extensive evaluations show that SIHD significantly outperforms state-of-the-art baselines in decision-making performance and demonstrates superior generalization across diverse scenarios.

## 1 Introduction

Offline reinforcement learning (also known as batch RL) [Lange et al., 2012] trains policies using pre-collected data without further interaction with the environment [Levine et al., 2020]. This paradigm is particularly well-suited to high-stakes domains where online data collection is costly or infeasible, such as healthcare [Tang et al., 2022], education [De Lima and Krohling, 2021], and robotic control [Villaflor et al., 2022]. Out-of-distribution (OOD) states and actions—underrepresented or absent in offline datasets—often cause temporal-difference learning methods to suffer from severe extrapolation errors [Kumar et al., 2019, Fujimoto et al., 2019]. Additionally, suboptimal and diverse historical trajectories give rise to the mode-covering challenge [Chen et al., 2023, 2024b], where conventional unimodal policies fail to capture the multi-modal nature of offline behavioral strategies.

To mitigate these challenges, recent approaches have incorporated diffusion models [Ho et al., 2020] with strong generative capacity to build expressive diffusion-based policies [Chen et al., 2023, 2024b]. Offline sequential decision-making has been further reframed as a trajectory generation task, where diffusion models are conditioned on reward-related signals (e.g., returns or reward-to-go) to generate high-return sequences [Liang et al., 2023, He et al., 2023]. However, deploying these models in

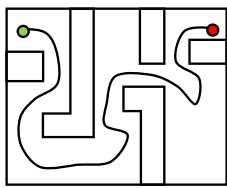 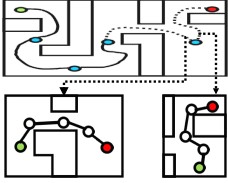 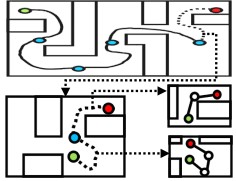 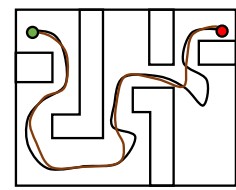

a) offline trajectory    b) 2-layer rigid diffusion    c) multi-scale hierarchy    d) regularized exploration

Figure 1: Illustrative example of navigation from the green start point to the red goal: (a) the offline suboptimal trajectory; (b) the rigid two-layer diffusion hierarchy with a single predefined temporal scale; (c) the adaptive multi-scale hierarchical diffusion framework based on structural information principles; (d) the enhanced state exploration guided by structural entropy regularization.

long-horizon tasks remains difficult due to the exponential increase in value estimate variance [Ren et al., 2021] and the high computational cost of iterative denoising steps [Wang et al., 2022].

To improve efficiency in long-horizon decision-making, hierarchical policies have been incorporated into diffusion-based offline RL, enabling the decomposition of complex tasks into manageable subproblems guided by intermediate subgoals [Sacerdoti, 1974, Pertsch et al., 2020]. Early approaches [Ajay et al., 2022, He et al., 2023] introduce manually predefined skills into diffusion models via one-hot encodings, which limit the scalability of the resulting policies. Subsequently, the HDMI framework [Li et al., 2023] advances this direction by learning reward-conditioned subgoals and generating goal-directed trajectories using a hierarchical diffusion model. Building on this, the Hierarchical Diffuser [Chen et al., 2024a] utilizes lightweight trajectory segmentation to streamline subgoal inference, further enhancing long-horizon planning performance.

Despite their success, existing methods typically rely on a single predefined temporal scale to segment offline trajectories (Figure 1(a)) and assume a fixed two-layer diffusion hierarchy composed of subgoal and action layers (Figure 1(b)). Such rigid structures hinder adaptability to varying temporal patterns and task-specific complexities, limiting both decision-making performance and flexibility. Recent work [Evans and Şimşek, 2023] has demonstrated the potential of multi-layer policy hierarchies, structured from state transition dynamics, to enhance generalization in compositional long-horizon tasks. This raises a central open challenge in offline RL: how can historical trajectories be systematically analyzed to construct a diffusion hierarchy that is both generalizable and task-aware?

In this work, we propose a novel **S**tructural **I**nformation-based **H**ierarchical **D**iffusion framework, called **SIHD**, for stable and effective offline policy learning. We begin by extracting structural information from a similarity-guided topological graph constructed over state elements, from which tree-structured state communities are derived. Offline trajectories are adaptively segmented based on community partitioning at each layer, enabling the construction of a multi-scale diffusion hierarchy (Figure 1(c)). We quantify the structural information gain of each state community and use it as conditional guidance between adjacent diffusion layers, thereby reducing reliance on reward prediction over locally receptive sub-trajectories. We further introduce a structural entropy regularizer to promote exploration of underrepresented states in historical trajectories (Figure 1(d)), mitigating over-reliance on offline datasets. To prevent extrapolation errors arising from distributional shifts between behavioral and learned policies, this exploration is constrained to the lowest-level communities in the hierarchy. Comparative evaluations on the D4RL benchmark show that SIHD outperforms both non-hierarchical and hierarchical state-of-the-art baselines by up to $12.6\%$ in decision-making performance and exhibits superior generalization in long-horizon, sparse-reward tasks. Our contributions are summarized as follows:

• We propose a novel hierarchical diffusion framework that leverages structural information from historical trajectories to enable adaptive modeling across multiple temporal scales.

• We quantify the structural information gain of each state community and use it as conditional guidance for the corresponding diffuser over localized sub-trajectories.

• We introduce a structural entropy regularizer that promotes exploration of underrepresented states within low-level communities, mitigating overreliance on offline datasets and reducing extrapolation errors from distributional shifts.

• We conduct comprehensive evaluations on the D4RL benchmark, demonstrating that SIHD significantly outperforms state-of-the-art baselines in both decision-making performance and generalization on long-horizon, sparse-reward tasks.

## 2 Related Work

### 2.1 Hierarchical Decision-Making

Hierarchical reinforcement learning (HRL) [Sacerdoti, 1974, Pertsch et al., 2020] addresses long-horizon decision-making tasks by decomposing them into layered subtasks, where higher-layer policies orchestrate lower-layer primitives to enable temporal abstraction. For example, Iris [Mandlekar et al., 2020] learns implicit hierarchical policies from offline robotic data, while HiGoC [Li et al., 2022] integrates hierarchical abstraction with goal-conditioned offline RL for multi-stage planning. Multi-level hierarchies [Evans and Şimşek, 2023] extend HRL by systematically organizing skills at varying levels of granularity, enabling flexible adaptation to compositional tasks.

In offline settings, HRL mitigates challenges such as distributional shift and sparse rewards through structured policy decomposition. OPAL [Ajay et al., 2021] accelerates learning by discovering reusable primitives from offline datasets, while recent work [Villecroze et al., 2022, Rao et al., 2022] explores data-driven skill extraction without predefined task hierarchies. Despite these advances, offline HRL remains prone to instability due to the deadly triad of function approximation, bootstrapping, and off-policy learning [Van Hasselt et al., 2018], further exacerbated by limited data coverage [Ma et al., 2022] and persistent reward sparsity [Kumar et al., 2020].

These issues underscore the need for more stable and generalizable hierarchical frameworks for offline policy learning in long-horizon decision-making scenarios with limited interaction and feedback from the environment.

### 2.2 Diffusion-based Offline RL

Diffusion models [Ho et al., 2020] have emerged as a powerful tool in offline reinforcement learning (RL), offering strong distribution-matching and sequence-generation capabilities that help mitigate extrapolation errors [Kumar et al., 2019, Fujimoto et al., 2019] and poor mode coverage [Chen et al., 2023, 2024b] commonly observed in conventional methods. By synthesizing high-fidelity trajectories aligned with offline data, diffusion-based methods [Janner et al., 2022] reformulate policy optimization as a generative modeling problem, enabling diverse behaviors while avoiding out-of-distribution actions.

Despite these strengths, diffusion-based offline RL faces challenges in long-horizon tasks, where variance accumulation across extended trajectories [Ren et al., 2021] and high computational costs [Wang et al., 2022] limit scalability. Recent work has sought to address these limitations by introducing hierarchical diffusion models [Li et al., 2023, Chen et al., 2024a], which decompose decision-making into a two-stage process: high-layer goal planning followed by low-layer action generation. However, these methods typically rely on fixed temporal segmentation and manually defined two-layer hierarchies, limiting their adaptability to tasks requiring dynamic horizon adjustments or reasoning over multiple temporal scales.

These limitations underscore the need for hierarchical diffusion frameworks that are flexible, task-aware, and adaptively constructed from offline trajectories to enable robust long-horizon decision-making in offline RL.

### 2.3 Structural Information Principles

Structural information principles [Li and Pan, 2016, Li, 2024] quantify uncertainty in graph-based dynamics through structural entropy and hierarchical partitioning. Specifically, structural entropy measures the minimum number of bits required to encode the probability distribution of vertices reachable via a single-step random walk on a graph. In its one-dimensional form, it corresponds to the Shannon entropy of the graph's degree distribution, providing an upper bound on dynamic uncertainty. Hierarchically grouping tightly connected vertices into communities reduces the encoding cost, yielding a tree-structured partitioning known as an encoding tree. Encoding trees derived from topological graphs have been effectively applied in graph learning [Wu et al., 2022], network analysis [Zeng et al., 2024], and decision-making tasks [Zeng et al., 2023a,b].

In this work, we leverage structural information embedded in historical trajectories to construct an adaptive multi-scale hierarchical diffusion framework, enabling stable and effective policy learning in long-horizon, sparse-reward environments.

# 3 Preliminaries

In this section, we formalize the foundational concepts of offline reinforcement learning (RL), diffusion probabilistic models, and structural information principles. The primary notations used throughout the paper are summarized in Appendix A.1 for reference.

## 3.1 Offline RL

Reinforcement learning is typically formalized as a Markov Decision Process (MDP) [Bellman, 1957], defined by the tuple $\mathcal{M} = <\mathcal{S}, \mathcal{A}, \mathcal{P}, \mathcal{R}, \gamma>$, where $\mathcal{S}$ denotes the state space, $\mathcal{A}$ the action space, $\mathcal{P}(s'|s,a)$ the transition function, $\mathcal{R}(s,a)$ the reward function, and $\gamma$ the discount factor. The objective of a parameterized policy $\pi_\theta(a|s)$ is to maximize the expected discounted return, $\mathbb{E}_{s_0=s, a_t \sim \pi_\theta(\cdot|s_t), s_{t+1} \sim \mathcal{P}(\cdot|s_t, a_t)} [\sum_{t=0}^{\infty} \gamma^t \mathcal{R}(s_t, a_t)]$. This return is estimated using a state-action value function $\mathcal{Q}_\phi(s,a)$, which is learned via temporal-difference methods [Sutton, 1988].

In offline settings, the policy $\pi_\theta(a|s)$ is trained exclusively on a static dataset $\mathcal{D}_{\pi_b} = \{(s, a, r, s')\}$ collected by a behavioral policy $\pi_b(a|s)$. The training objective of offline RL [Wu et al., 2019] regularizes the learned policy $\pi_\theta$ towards the policy $\pi_b$ and is reformulated as follows:

$$\max_{\pi_\theta} \mathbb{E}_{s \sim \mathcal{D}_{\pi_b}, a \sim \pi_\theta(\cdot|s)} \left[ \mathcal{Q}_\phi(s, a) - \eta D_{KL} \left[ \pi_\theta(\cdot|s) || \pi_b(\cdot|s) \right] \right], \tag{1}$$

where $\eta$ is the temperature coefficient that balances exploitation and behavioral regularization.

## 3.2 Diffusion Probabilistic Models

Diffusion probabilistic models [Ho et al., 2020] are generative methods that progressively corrupt data samples with noise until they resemble a standard Gaussian distribution, and train a model to reverse this transformation. The forward process gradually adds Gaussian noise $\epsilon \sim \mathcal{N}(0, \mathcal{I})$ to data samples $x_0$ over $K$ steps, according to a predefined variance schedule $\beta_1, \cdots, \beta_K$:

$$q(x_{1:K}|x_0) = \prod_{k=1}^{K} q(x_k|x_{k-1}), \quad q(x_k|x_{k-1}) = \mathcal{N}(x_k; \sqrt{1-\beta_k} x_{k-1}, \beta_k \mathcal{I}). \tag{2}$$

To recover the original data, the reverse process is trained to approximate the reverse transitions $q(x_{k-1} \mid x_k)$ by predicting the added noise using a neural network $\epsilon_\theta$, starting from $x_K \sim \mathcal{N}(0, \mathcal{I})$:

$$p_\theta(x_{0:K}) = p(x_K) \prod_{k=1}^{K} p_\theta(x_{k-1}|x_k), \quad p_\theta(x_{k-1}|x_k) = \mathcal{N}(x_{k-1}; \mu_\theta(x_k, k), \Sigma_\theta(x_k, k)), \tag{3}$$

where $\mu_\theta$ and $\Sigma_\theta$ denote the mean and covariance of the predicted Gaussian distribution.

Building on diffusion probabilistic models, the offline decision-making task is framed as a goal-conditional generative modeling problem [He et al., 2023, Li et al., 2023] by maximizing:

$$\mathbb{E}_{\tau_0 \sim \mathcal{D}_{\pi_b}} \left[ \log p_\theta(\tau_0|y(\tau_0)) \right], \quad \tau_0 = \begin{bmatrix} s_0 & s_1 & \cdots & s_T \\ a_0 & a_1 & \cdots & a_T \end{bmatrix}, \tag{4}$$

where $\tau_0$ is a historical trajectory sampled from $\mathcal{D}_{\pi_b}$, and $y(\tau_0)$ is the conditional information (e.g., target cumulative reward or constraint satisfaction).

## 3.3 Structural Information Principles

In structural information principles [Li and Pan, 2016], the encoding tree $\mathcal{T}$ of an undirected graph $\mathcal{G} = (\mathcal{V}, \mathcal{E})$ satisfies the following properties: 1) Each node $\alpha$ in $\mathcal{T}$ uniquely corresponds to a subset $\mathcal{V}_\alpha \subseteq \mathcal{V}$. 2) The root node $\lambda$ satisfies $\mathcal{V}_\lambda = \mathcal{V}$. 3) For each leaf node $\nu$ in $\mathcal{T}$, $\mathcal{V}_\nu$ is a singleton $\{v\}$ for some $v \in \mathcal{V}$. 4) For every non-leaf node $\alpha$ with $l_\alpha$ children $\alpha_1, \cdots, \alpha_{l_\alpha}$, the subsets $\{\mathcal{V}_{\alpha_i}\}_{i=1}^{l_\alpha}$ form a partitioning of $\mathcal{V}_\alpha$, i.e., $\mathcal{V}_\alpha = \bigcup_{i=1}^{l_\alpha} \mathcal{V}_{\alpha_i}$ and $\mathcal{V}_{\alpha_i} \cap \mathcal{V}_{\alpha_j} = \emptyset$ for $i \neq j$.

The $\mathcal{K}$-dimensional structural entropy of $\mathcal{G}$ is defined over all encoding trees $\mathcal{T}$ of height $h_\mathcal{T} \leq \mathcal{K}$ as:

$$\mathcal{H}^{\mathcal{K}}(\mathcal{G}) = \min_{h_\mathcal{T} \leq \mathcal{K}} \mathcal{H}^{\mathcal{T}}(\mathcal{G}), \quad \mathcal{H}^{\mathcal{T}}(\mathcal{G}) = - \sum_{\alpha \in \mathcal{T}, \alpha \neq \lambda} \left[ \frac{g_\alpha}{\text{vol}(\lambda)} \cdot \log \frac{\text{vol}(\alpha)}{\text{vol}(\alpha^-)} \right], \tag{5}$$

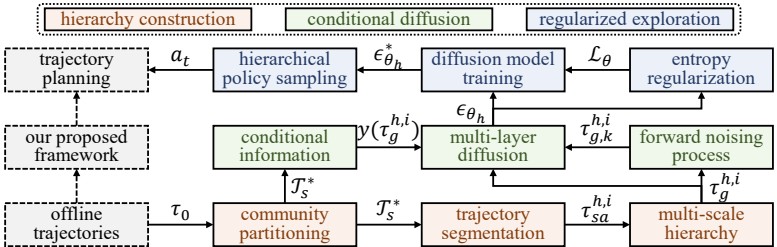

Figure 2: The proposed SIHD framework comprising the hierarchy construction, conditional diffusion, and regularized exploration modules.

where $\alpha^-$ denotes the parent of a non-root node $\alpha$, $\text{vol}(\alpha) = \sum_{v \in \mathcal{V}_\alpha} d_v$ is the sum of degrees (volume) of the vertex subset $\mathcal{V}\alpha$, and $g_\alpha$ is the total weight of edges that cross the cut between $\mathcal{V}\alpha$ and its complement $\mathcal{V} \setminus \mathcal{V}\alpha$.

# 4 The SIHD Framework

In this section, we present the detailed design of the proposed SIHD framework, whose overall architecture is illustrated in Figure 2. Superscripts $h$ and $i$ denote the diffusion layer and the temporal index within the same layer, respectively, while subscripts $g$ and $sa$ indicate subgoal sequences and state-action subtrajectories.

The hierarchy construction module models the topological structure of offline states based on feature similarity and identifies hierarchical communities by optimizing structural entropy to construct an adaptive multi-scale diffusion hierarchy (see Section 4.1). The conditional diffusion module feeds temporally segmented trajectories into corresponding layers of a shared diffusion model, integrating quantized rewards and structural signals to perform forward diffusion and reverse inference at multiple time scales (see Section 4.2). The regularized exploration module employs structural entropy as a measure of generative diversity and adds a regularization loss that encourages exploration of underrepresented offline states during both training and inference (see Section 4.3).

## 4.1 Multi-Scale Diffusion Hierarchy

Instead of the rigid two-layer diffusion hierarchy operating at a single scale [Li et al., 2023, Chen et al., 2024a], we leverage feature similarity to identify structural relationships among offline states and derive a tree-structured partitioning of state communities to construct the trajectory-adaptive multi-scale hierarchical diffusion framework.

Starting from historical trajectories $\mathcal{D}_{\pi_b}$ (Figure 3(a)), we extract all state elements $s \in \mathcal{S}$ to form the vertex set $S$ of observed states and establish weighted edges based on feature similarity (e.g., cosine similarity) between each vertex and its top-k nearest neighbors, thereby forming the k-nearest-neighbor state graph $\mathcal{G}_s$. We determine the parameter k by selecting the value that maximizes the upper bound of dynamic uncertainty, $\mathcal{H}^1(\mathcal{G}_s)$, to promote structural expressiveness for encoding trees over $\mathcal{G}_s$. We then apply the HCSE optimization algorithm Pan et al. [2025] to derive the height-$\mathcal{K}$ optimal encoding tree $\mathcal{T}_s^*$ that minimizes the $\mathcal{K}$-dimensional structural entropy $\mathcal{H}^{\mathcal{K}}(\mathcal{G}_s)$. The optimization procedure is detailed in Appendix A.2. As discussed in Section 3.3, the encoding tree $\mathcal{T}_s^*$ represents a tree-structured partitioning of $\mathcal{G}_s$ into communities (Figure 3(b)), where each node $\alpha$ corresponds to a state community $\mathcal{V}_\alpha \subseteq S$ at a particular level of granularity, and the parent-child relationships reflect the hierarchical inclusion structure among communities.

Based on the community partitioning defined by $\mathcal{T}_s^*$, we perform adaptive hierarchical segmentation for each trajectory $\tau_0$ in $\mathcal{D}_{\pi_b}$ (Figure 3(c)). Specifically, for each layer $h$, we obtain the set of nodes $\mathcal{U}_h$ at height $h$ from $\mathcal{T}_s^*$. We then segment the trajectory $\tau_0$ according to the community assignment defined by $\mathcal{U}_h$, ensuring each resulting segment $\tau_{sa}^{h,i}$ is a temporally continuous sequence of states from the same community. Each trajectory segment $\tau_{sa}^{h,i}$ is formally defined as follows:

$$\tau_{sa}^{h,i} = \begin{bmatrix} s_{\sum_{j=1}^{i-1} l_{sa}^{h,j}+1} & \cdots & s_{\sum_{j=1}^{i} l_{sa}^{h,j}} \\ a_{\sum_{j=1}^{i-1} l_{sa}^{h,j}+1} & \cdots & a_{\sum_{j=1}^{i} l_{sa}^{h,j}} \end{bmatrix}, \quad g_i^h = s_{\sum_{j=1}^{i} l_{sa}^{h,j}}, \tag{6}$$

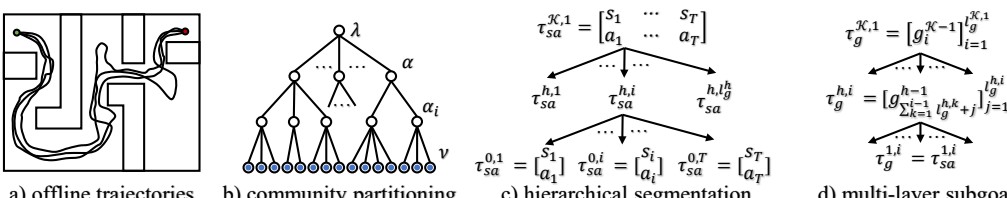

a) offline trajectories    b) community partitioning    c) hierarchical segmentation    d) multi-layer subgoals

Figure 3: Construction of the multi-scale hierarchical diffusion framework by minimizing the structural entropy of offline trajectories, deriving tree-structured community partitioning, segmenting trajectories hierarchically, and extracting multi-layer subgoal sequences.

where $i$ represents the temporal index of the $i$-th segment at layer $h$, and the final state in $\tau_{sa}^{h,i}$ is designated as the subgoal $g_i^h$. The variable $l_{sa}^{h,i}$ denotes the sequence length (i.e., temporal receptive field) of the segment $\tau_{sa}^{h,i}$. The complete segmentation procedure is detailed in Appendix A.4.

We integrate the $\mathcal{K}$-layer subgoals, extracted from the segmentation hierarchy (Figure 3(d)), into the control-as-inference framework [Levine, 2018] to define a hierarchical, multi-scale diffusion framework. At each layer $h$, we define a subgoal sequence $\tau_g^h$ of length $l_g^h$, where for each subgoal $g_i^h$, the corresponding subgoal sequence $\tau_g^{h,i} \subseteq \tau_g^{h-1}$ at the next lower layer is defined recursively as:

$$\tau_g^{h,i} = \begin{cases} \left[ g_{\sum_{j=1}^{i-1} l_g^{h,j}+1}^{h-1} \quad \cdots \quad g_{\sum_{j=1}^{i} l_g^{h,j}}^{h-1} \right] & \text{if } h > 1, \\ \tau_{sa}^{h,i} & \text{if } h = 1. \end{cases} \tag{7}$$

The temporal scale $l_g^{h,i}$ of subgoal $g_i^h$ is not predefined but inferred from the hierarchical partitioning in $\mathcal{T}_s^*$ and the reference trajectory $\tau_0$. For the base case where $h = 1$, the lower-layer subgoal sequence $\tau_g^{h,i}$ corresponds to the state-action segment $\tau_{sa}^{h,i}$.

Following the control-as-inference paradigm, we introduce a binary optimality variable $\mathcal{O}_i$ indicating whether the trajectory segment $\tau_{sa}^{\mathcal{K}-1,i}$ is optimal with respect to the subgoal $g_i^{\mathcal{K}-1}$. The posterior probability of $\mathcal{O}_i$ is modeled using a Boltzmann distribution over the cumulative reward:

$$p(\mathcal{O}_i = 1 | g_i^{\mathcal{K}-1}, \tau_{sa}^{\mathcal{K}-1,i}) \propto \exp \left( \sum_{k=1}^{l_{sa}^{\mathcal{K}-1,i}} \mathcal{R} \left( s_{\sum_{j=1}^{i-1} l_{sa}^{\mathcal{K}-1,j}+k}, a_{\sum_{j=1}^{i-1} l_{sa}^{\mathcal{K}-1,j}+k} \right) \right). \tag{8}$$

The following theorem enables the decomposition of the conditional generation problem for offline trajectories into a hierarchical diffusion process with dynamic temporal scales across multiple layers. The detailed proof is provided in Appendix B.1.

**Theorem 4.1.** *Given the $\mathcal{K}$-layer subgoals for each offline trajectory $\tau_0$, constructed as described above, the conditional probability in Equation 4 can be factorized as follows:*

$$p(\tau_0 | y(\tau_0)) \propto p(\tau_g^{\mathcal{K},1}) y(\tau_g^{\mathcal{K},1}) \cdot \prod_{h=1}^{\mathcal{K}-1} \prod_{i=1}^{l_g^h} p(\tau_g^{h,i}) y(\tau_g^{h,i}), \tag{9}$$

*where $y(\tau_g^{\mathcal{K},1}) = \exp \left( \sum_{t=0}^{T} \mathcal{R}(s_t, a_t) \right)$ captures the cumulative reward over the full trajectory $\tau_0$, $y(\tau_g^{h,i})$ imposes subgoal satisfaction constraints via a Dirac delta function, ensuring exact compliance, and $l_g^h$ denotes the length of the subgoal sequence $\tau_g^h$ at the $h$-th layer.*

## 4.2 Conditional Diffusion Model

At the top of the hierarchy, the diffusion model generates a subgoal sequence conditioned on the overall task reward. Subsequent layers generate subgoal sequences at intermediate levels of the hierarchy or state-action sequences at the base level, with each generation process conditioned on the parent subgoal. At each of these layers, we compute the structural information gain within the associated state communities, integrating it into classifier-free guidance to enhance conditional sequence generation.

For the $h$-th diffuser, we denote each target sequence $\tau_g^{h,i}$ (a subgoal sequence if $h > 1$, or a state-action sequence if $h = 1$) as the input $\tau_{g,0}^{h,i}$ to the diffusion process. We then iteratively construct a

sequence of progressively noised samples $\tau_{g,k}^{h,i}$ via the forward diffusion process:

$$q(\tau_{g,k}^{h,i}|\tau_{g,k-1}^{h,i}) = \mathcal{N}(\tau_{g,k}^{h,i}; \sqrt{1-\beta_k}\tau_{g,k-1}^{h,i}, \beta_k \mathcal{I}), \quad 1 \le i \le l_g^h. \tag{10}$$

To reduce training overload, we adopt a shared diffusion model $\epsilon_{\theta_h}$ for layer $h$. This model operates with classifier-free guidance [Liu et al., 2023] to jointly predict the noise term $\epsilon \sim \mathcal{N}(0, \mathcal{I})$ and estimate the reverse-step posterior distribution $p_{\theta_h}(\tau_{g,k-1}^{h,i} \mid \tau_{g,k}^{h,i})$ at each diffusion step $k$:

$$p_{\theta_h}(\tau_{g,k-1}^{h,i}|\tau_{g,k}^{h,i}) = \mathcal{N}(\tau_{g,k-1}^{h,i}; \mu_{\theta_h}, \Sigma_{\theta_h}), \quad \hat{\epsilon} = \epsilon_{\theta_h}(\tau_{g,k}^{h,i}, (1-\omega)y(\tau_{g,k}^{h,i}) + \omega\emptyset, k), \tag{11}$$

where $\mu_{\theta_h}$ and $\Sigma_{\theta_h}$ denote the mean vector and covariance matrix of the estimated noise $\hat{\epsilon}$. The symbol $\emptyset$ represents the absence of conditional input and replaces $y(\tau_{g,k}^{h,i})$ during unconditional generation with a guidance weight $\omega$.

As specified in Theorem 4.2, the conditional input to the highest-level diffusion model, $y(\tau_{g,k}^{\mathcal{K},1})$, is defined as the exponential of the cumulative reward over the full trajectory $\tau_0$. At lower layers, we use structural information-based values instead of cumulative rewards. For each sequence $\tau_g^{h,i}$, we identify the tree node $\alpha$ at height $h$ in the encoding tree $\mathcal{T}_s^*$ such that all states in the corresponding state-action segment $\tau_{sa}^{h,i}$ belong to the community $\mathcal{V}_\alpha$. We then define the conditional input $y(\tau_g^{h,i})$ as the structural information gain of node $\alpha$:

$$y(\tau_g^{h,i}) = \mathcal{H}^{\mathcal{T}_s^*}(\mathcal{G}_s; \alpha) = -\frac{g_\alpha}{\text{vol}(\lambda)} \cdot \log \frac{\text{vol}(\alpha)}{\text{vol}(\alpha^-)}. \tag{12}$$

This gain term quantifies the additional information required to infer that a single-step random state transition in $\mathcal{G}_s$ occurs within the lower-level segment $\tau_{sa}^{h,i}$, given prior knowledge, provided by the subgoal $g_i^h$, that the transition is contained within the higher-level segment $\tau_{sa}^{h+1,j}$. To ensure consistency with the hierarchical subgoal constraints defined in Theorem 4.2, we replace the terminal state of each denoised trajectory $\hat{\tau}_{g,0}^{h,i}$ with its corresponding subgoal $g_i^h$.

## 4.3 Structural Entropy Regularizer

To mitigate overreliance on the limited state coverage of offline datasets, we introduce a structural entropy-based exploration regularizer that encourages the hierarchical diffusion policy to explore underrepresented regions of the state space while limiting extrapolation errors resulting from deviations from the offline behavioral policy.

For each trajectory $\tau_0$, we extract the state transition $(s_t, s_{t+1})$ at each timestep $t$ and estimate the transition probability $p_{\theta_1}(s_{t+1}, s_t)$ using the base-layer diffusion model $\epsilon_{\theta_1}$. These estimated transition probabilities are used in place of feature similarity metrics to construct a new topology over the state set $S$, resulting in a complete weighted state graph $\mathcal{G}_s'$, where each edge weight reflects diffusion-inferred transition likelihood. In $\mathcal{G}_s'$, the weighted degree of each state $s \in S$ is defined as its diffusion-based visitation probability $p_{\theta_1}(s)$, computed by summing the incoming transition probabilities. We define the structural entropy of the graph $\mathcal{G}_s'$ under the encoding tree $\mathcal{T}_s^*$ as:

$$\mathcal{H}^{\mathcal{T}_s^*}(\mathcal{G}_s') = -\sum_{\alpha \in \mathcal{T}_s^*, \alpha \neq \lambda} \left[ \frac{\sum_{s_i \notin \mathcal{V}_\alpha} \sum_{s_j \in \mathcal{V}_\alpha} p_{\theta_1}(s_i, s_j)}{\sum_{s \in \mathcal{V}} p_{\theta_1}(s)} \cdot \log \frac{\sum_{s \in \mathcal{V}_\alpha} p_{\theta_1}(s)}{\sum_{s \in \mathcal{V}_{\alpha^-}} p_{\theta_1}(s)} \right]. \tag{13}$$

The following theorem provides a variational lower bound on $\mathcal{H}^{\mathcal{T}_s^*}(\mathcal{G}_s')$ and establishes its theoretical connection to the Shannon entropy of the state distribution $p_{\theta_1}$ over $S$. A detailed proof is provided in Appendix B.2.

**Theorem 4.2.** *For the encoding tree $\mathcal{T}_s^*$ and the complete weighted state graph $\mathcal{G}_s'$, the structural entropy $\mathcal{H}^{\mathcal{T}_s^*}(\mathcal{G}_s')$ admits the variational lower bound related to the Shannon entropy $\mathcal{H}(S)$:*

$$\mathcal{H}(S) - \sum_{h=1}^{\mathcal{K}-1} [\eta_h \cdot \mathcal{H}(\mathcal{U}_h)] \le \mathcal{H}^{\mathcal{T}_s^*}(\mathcal{G}_s') \le \mathcal{H}(S), \tag{14}$$

*where each layer-specific weight $\eta_h$ is defined as the maximum of $\frac{\sum_{i=1}^{l_\alpha} g_{\alpha_i} - g_\alpha}{\text{vol}(\alpha)}$ over all nodes $\alpha$ at height $h$ in $\mathcal{T}_s^*$ for the state graph $\mathcal{G}_s'$.*

---
**Algorithm 1** The SIHD Planning Algorithm

---
1: **Input:** hierarchical diffusion probabilistic models $\{\epsilon_{\theta_h}\}_{h=1}^{\mathcal{K}}$, the initial state $s_0 \in \mathcal{S}$, the planning horizon $H$, the maximal cumulative reward $r_{max}$ in $\mathcal{D}_{\pi_b}$
2: **Output:** the action sequence $\{a_t\}_{t=0}^{H-1}$
3: Initialize the hierarchical subgoal sequence $\tau_g^{h,1} \leftarrow [s_0]$ for $2 \leq h \leq \mathcal{K}$
4: Initialize the state-action sequence $\tau_g^{1,1}[0,:] = [s_0]$ and $\tau_g^{1,1}[1,:] = [\emptyset]$ at the 1-th layer
5: **for** $t \leftarrow 0$ to $H - 1$ **do**
6:     Sampling the starting noised sequence $\hat{\tau}_g^{1,1} \sim \mathcal{N}(0, \mathcal{I})$
7:     $\hat{\tau}_{g,k}^{1,1}[:,:l_g^{1,1}] \leftarrow \tau_g^{1,1}$
8:     **if** $\tau_g^{2,1}[-1]$ is satisfied **then**
9:         $\tau_g^{2,1} \leftarrow f_{su}(2, \{\epsilon_{\theta_h}\}_{h=1}^{\mathcal{K}}, \{\tau_g^{h,1}\}_{h=1}^{\mathcal{K}}, r_{max})$
10:     **end if**
11:     **for** $k \leftarrow K$ to $1$ **do**
12:         $\alpha \leftarrow$ select the 1-layer node according to $\tau_g^{2,1}[-1]$
13:         Calculate the conditional information $y(\hat{\tau}_{g,k}^{1,1})$ via Equation 12
14:         $\hat{\epsilon} \leftarrow \epsilon_{\theta_1}(\hat{\tau}_{g,k}^{1,1}, (1 - \omega)y(\hat{\tau}_{g,k}^{1,1}) + \omega\emptyset, k)$
15:         $(\mu_{\theta_1}, \Sigma_{\theta_1}) \leftarrow$ extract the mean vector and covariance matrix from $\hat{\epsilon}$
16:         $\hat{\tau}_{g,k-1}^{1,1} \sim \mathcal{N}(\mu_{\theta_1}, \beta_k \Sigma_{\theta_1})$
17:     **end for**
18:     $\tau_g^{1,1}[0,:] \leftarrow \tau_g^{1,1}[0,:] + \hat{\tau}_{g,0}^{1,1}[0, l_g^{1,1}]$
19:     $\tau_g^{1,1}[1,:] \leftarrow \tau_g^{1,1}[1,:l_g^{1,1} - 1] + [\hat{\tau}_{g,0}^{1,1}[1, l_g^{1,1}], \emptyset]$
20: **end for**
21: Return the sequence $\tau_g^{1,1}[1,:]$ as the action sequence $\{a_t\}_{t=0}^{H-1}$

---

To promote more balanced coverage of the state space, we maximize the Shannon entropy $\mathcal{H}(S)$ over the state distribution, thereby improving exploration of underrepresented states and reducing dependency on historical trajectories. Conversely, to preserve the decision-making hierarchy encoded in $\mathcal{T}_s^*$, we minimize the structural entropy $\mathcal{H}(\mathcal{U}_h)$ (structural entropy of the community partition at each layer $h$), with appropriate weighting $\eta_h$. This regularization discourages large deviations from the behavioral policy $\pi_b$ and mitigates extrapolation errors.

The regularized training objective for each diffusion model $\epsilon_{\theta_h}$ is defined as:

$$\mathcal{L}(\theta_h) = \mathbb{E} \sum_{i=1}^{l_g^h} \left[ ||\epsilon_{\theta_h}(\tau_{g,k}^{h,i}, (1 - \omega)y(\tau_{g,k}^{h,i}) + \omega\emptyset, k) - \epsilon||^2 - \eta\mathbb{I}_{h=1} \left[ \mathcal{H}(S) - \sum_{j=1}^{\mathcal{K}-1} [\eta_j \cdot \mathcal{H}(\mathcal{U}_j)] \right] \right],$$
(15)

where $\mathbb{I}_{h=1}$ is an indicator function equal to 1 if $h = 1$, and $\eta$ is a weighting coefficient for the entropy regularization term. The training procedure of the SIHD framework is summarized in Appendix A.5.

### 4.4 The SIHD Planning

During inference, our SIHD planner proceeds as follows: We first initialize each hierarchy's subgoal buffers and the one-layer state–action sequence (lines 3 and 4 in Algorithm 1). For each planning step, we (i) sample an initial noisy latent sequence from the diffuser prior (lines 6 and 7 in Algorithm 1), (ii) invoke the subgoal proposer to revise the top-level goal if its terminal criterion is satisfied (lines 8–10 in Algorithm 1), and (iii) execute a reverse-diffusion rollout across latent timesteps, where at each diffuser step we sample the next latent sequence (lines 11–17 in Algorithm 1). Finally, we decode and update the one-layer state–action sequence (lines 18 and 19 in Algorithm 1), and after $H$ iterations, we output the action trajectory (line 21 in Algorithm 1).

## 5 Experiment

In this section, we conduct comparative experiments on the D4RL benchmark [Fu et al., 2020], covering standardized offline and long-horizon planning tasks, to evaluate decision-making performance

Table 1: Performance comparison between SIHD and baseline methods on the Medium-Expert, Medium, and Medium-Replay datasets from the D4RL Gym-MuJoCo benchmark tasks: "average reward $\pm$ standard deviation". **Bold**: the best performance, underline: the second performance.

| Gym-MuJoCo | HalfCheetah | | | Hopper | | | Walker2D | | |
|---|---|---|---|---|---|---|---|---|---|
| Tasks | Expert | Medium | Replay | Expert | Medium | Replay | Expert | Medium | Replay |
| CQL | 91.6 | 44.0 | 45.5 | 105.4 | 58.5 | 95.0 | 108.8 | 72.5 | 77.2 |
| IQL | 86.7 | 47.4 | 44.2 | 91.5 | 66.3 | 94.7 | 109.6 | 78.3 | 73.9 |
| DT | 86.8 | 42.6 | 36.6 | 107.6 | 67.6 | 82.7 | 108.1 | 74.0 | 66.6 |
| MoReL | 53.3 | 42.1 | 40.2 | 108.7 | 95.4 | 93.6 | 95.6 | 77.8 | 49.8 |
| TT | 95.0 | 46.9 | 41.9 | 110.0 | 61.1 | 91.5 | 101.9 | 79.0 | 82.6 |
| Diffuser | $88.9 \pm 0.3$ | $42.8 \pm 0.3$ | $37.7 \pm 0.5$ | $103.3 \pm 1.3$ | $74.3 \pm 1.4$ | $93.6 \pm 0.4$ | $106.9 \pm 0.2$ | $79.6 \pm 0.6$ | $70.6 \pm 1.6$ |
| HDMI | $92.1 \pm 1.4$ | $\underline{48.0} \pm 0.9$ | $44.9 \pm 2.0$ | $113.5 \pm 0.9$ | $76.4 \pm 2.6$ | $\underline{99.6} \pm 1.5$ | $107.9 \pm 1.2$ | $79.9 \pm 1.8$ | $80.7 \pm 2.1$ |
| HD | $\underline{92.5} \pm 0.3$ | $46.7 \pm 0.2$ | $38.1 \pm 0.7$ | $\underline{115.3} \pm 1.1$ | $\underline{99.3} \pm 0.3$ | $94.7 \pm 0.7$ | $107.1 \pm 0.1$ | $\underline{84.0} \pm 0.6$ | $\underline{84.1} \pm 2.2$ |
| SIHD | $\mathbf{94.4} \pm 0.5$ | $\mathbf{48.7} \pm 1.1$ | $\mathbf{47.0} \pm 0.4$ | $\mathbf{117.7} \pm 1.4$ | $\mathbf{103.1} \pm 0.8$ | $\mathbf{101.5} \pm 1.3$ | $\mathbf{110.3} \pm 0.9$ | $\mathbf{88.3} \pm 0.9$ | $\mathbf{89.7} \pm 1.9$ |
| Abs.(%) Avg. | 1.9(2.1) | 0.7(1.5) | 1.5(3.3) | 2.4(2.1) | 3.8(3.8) | 1.9(1.9) | 0.7(0.6) | 4.3(5.1) | 5.6(6.7) |

Table 2: Performance comparison between SIHD and baseline methods on the U-Maze, Medium, and Large datasets from the Maze2D and AntMaze tasks: "average reward $\pm$ standard deviation". **Bold**: the best performance, underline: the second performance.

| Gym-MuJoCo | Single-task Maze2D | | | Multi-task Maze2D | | | AntMaze | | |
|---|---|---|---|---|---|---|---|---|---|
| Tasks | U-Maze | Medium | Large | U-Maze | Medium | Large | U-Maze | Medium | Large |
| IQL | 47.4 | 34.9 | 58.6 | 24.8 | 12.1 | 13.9 | 62.2 | 70.0 | 47.5 |
| MPPI | 33.2 | 10.2 | 5.1 | 41.2 | 15.4 | 8.0 | - | - | - |
| Diffuser | $113.9 \pm 3.1$ | $121.5 \pm 2.7$ | $123.0 \pm 6.4$ | $128.9 \pm 1.8$ | $127.2 \pm 3.4$ | $132.1 \pm 5.8$ | $76.0 \pm 7.6$ | $31.9 \pm 5.1$ | - |
| IRIS | - | - | - | - | - | - | $89.4 \pm 2.4$ | $64.8 \pm 2.6$ | $43.7 \pm 1.3$ |
| HiGoC | - | - | - | - | - | - | $91.2 \pm 1.9$ | $79.3 \pm 2.5$ | $67.3 \pm 3.1$ |
| HDMI | $120.1 \pm 2.5$ | $121.8 \pm 1.6$ | $128.6 \pm 2.9$ | $131.3 \pm 1.8$ | $131.6 \pm 1.9$ | $135.4 \pm 2.5$ | - | - | - |
| HD | $\underline{128.4} \pm 3.6$ | $\underline{135.6} \pm 3.0$ | $\underline{155.8} \pm 2.5$ | $\underline{144.1} \pm 1.2$ | $\underline{140.2} \pm 1.6$ | $\underline{165.5} \pm 0.6$ | $\underline{94.0} \pm 4.9$ | $\underline{88.7} \pm 8.1$ | $\underline{83.6} \pm 5.8$ |
| SIHD | $\mathbf{144.6} \pm 2.9$ | $\mathbf{148.5} \pm 2.6$ | $\mathbf{161.7} \pm 3.4$ | $\mathbf{157.0} \pm 0.6$ | $\mathbf{156.8} \pm 1.7$ | $\mathbf{169.4} \pm 2.7$ | $\mathbf{96.5} \pm 2.8$ | $\mathbf{92.2} \pm 5.0$ | $\mathbf{89.4} \pm 4.2$ |
| Abs.(%) Avg. | 16.2(12.6) | 12.9(9.5) | 5.9(3.8) | 12.9(9.0) | 16.6(11.8) | 3.9(2.4) | 2.5(2.7) | 3.5(3.9) | 5.8(6.9) |

and generalization capabilities of the SIHD framework. The source code is publicly available via an anonymized link for peer review[1]. All experiments are run with five random seeds. Additional analyses, including hyperparameter sensitivity and visualization of model behavior, are provided in Appendix C.

## 5.1 Offline Reinforcement Learning

We begin by evaluating the SIHD framework, configured with $\mathcal{K} = 3$ diffusion layers, on standardized Gym-MuJoCo tasks, characterized by dense rewards and short horizons. This setup allows us to assess SIHD's offline decision-making capabilities across datasets of varying quality. For comparison, we consider both non-hierarchical and hierarchical state-of-the-art baselines. These include model-free methods such as CQL [Kumar et al., 2020], IQL [Kostrikov et al., 2022], and Decision Transformer (DT) [Chen et al., 2023]; model-based methods such as MoReL [Kidambi et al., 2020] and Trajectory Transformer (TT) [Janner et al., 2021]; and diffusion-based methods such as Diffuser [Janner et al., 2022], HDMI [Li et al., 2023], and Hierarchical Diffuser (HD) [Chen et al., 2024a].

As shown in Table 1, SIHD consistently achieves the highest average reward on offline datasets of varying quality for each task, demonstrating strong decision-making performance and generalization ability. Although hierarchical diffusion baselines such as HDMI and HD perform competitively on most datasets, they show degraded performance on others (e.g., Medium-Replay HalfCheetah and Medium-Expert Walker2d), likely due to their simplistic two-layer hierarchies and fixed single temporal scales. These results underscore the importance of incorporating an additional diffusion layer and adaptively inferred temporal scales from offline trajectories to enhance hierarchical diffusion performance. On the high-quality Medium-Expert dataset, SIHD achieves an average performance improvement of $1.6\%$ over the baselines. On the Medium and Medium-Replay datasets, which reflect average data quality, the performance advantage of SIHD is more pronounced, achieving performance improvements of $3.8\%$ and $3.9\%$, respectively. These improvements demonstrate that SIHD alleviates dependence on high-quality offline datasets by introducing structural entropy-regularized exploration, thereby enabling more effective offline policy learning.

## 5.2 Long-Horizon Planning

We next evaluate the long-horizon decision-making performance of SIHD, configured with an increased number of diffusion layers ($\mathcal{K} = 4$), on two sparse-reward navigation tasks: Maze2D

---
[1] https://github.com/SELGroup/SIHD.git

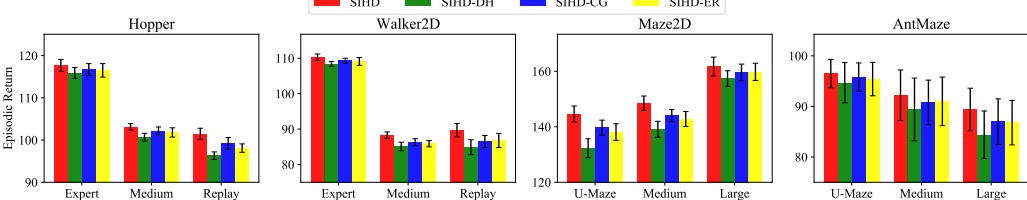

Figure 4: Ablation study on hierarchical construction, conditional diffusion, and regularized exploration within the SIHD framework, evaluated in the D4RL benchmark.

and AntMaze. In these environments, the agent receives a positive reward only upon successfully reaching a target location within an episode. The high-dimensional and noisy observations in these environments further exacerbate the decision-making challenges. In addition to the baselines used for Gym-MuJoCo, we include leading methods—MPPI [Williams et al., 2016], IRIS [Mandlekar et al., 2020], and HiGoC [Li et al., 2022]—as additional baselines for Maze2D and AntMaze. Following prior studies [Janner et al., 2022, Chen et al., 2024a], we introduce a multi-task variant of Maze2D in which the target location is randomized in each episode.

As shown in Table 2, SIHD achieves superior decision-making performance across all datasets, consistently attaining the highest task rewards among compared methods. Specifically, SIHD achieves average improvements in reward of $8.3\%$, $7.4\%$, and $4.4\%$ in the single-task Maze2D, multi-task Maze2D, and AntMaze tasks, respectively. Compared to the results on Gym-MuJoCo tasks, SIHD demonstrates a more pronounced performance advantage in navigation tasks, attributable to its deeper diffusion hierarchy, which better supports the long-term decision-making demands of these environments. Furthermore, SIHD demonstrates more stable performance across varying data qualities. For example, while the second-best performing HD baseline suffers maximum performance drops of $27.4$, $25.3$, and $10.4$ in the three task settings, SIHD limits these drops to just $17.1$, $12.6$, and $7.1$, respectively. This robustness stems from the structural entropy regularizer, which encourages exploration of underrepresented states, reduces reliance on historical trajectories, and thereby enhances generalization ability.

### 5.3 Ablation Study

To assess the contributions of key SIHD modules, we perform ablation studies on two Gym-MuJoCo tasks (Hopper and Walker2d) and two single-task navigation environments (Maze2D and AntMaze). Three model variants are constructed by selectively disabling core components: (1) replacing the multi-scale hierarchy (see Section 4.1) with a rigid two-layer single-scale diffuser (SIHD-DH); (2) removing the classifier-based guidance (see Section 4.2) (SIHD-CG); and (3) removing the structural entropy regularizer (see Section 4.3) (SIHD-ER). As shown in Figure 4, the full SIHD framework consistently outperforms all three ablated variants, highlighting the importance of each component. Notably, SIHD-DH shows a larger performance drop than SIHD-CG and SIHD-ER, underscoring the critical role of the multi-scale diffusion hierarchy. This effect becomes even more pronounced in long-horizon decision-making tasks. Additional experiments on sensitivity analysis, computational efficiency, extended ablation studies, and qualitative comparison are provided in Appendix C.

## 6 Conclusion

This work proposes SIHD, a novel hierarchical diffusion framework that leverages structural information embedded in historical trajectories to construct an adaptive multi-scale diffusion hierarchy and promote exploration of underrepresented states in offline datasets, thereby enabling effective policy learning. Extensive evaluations on the challenging D4RL benchmark demonstrate the superior decision-making performance and generalization capabilities of SIHD across diverse offline RL tasks. In future work, we aim to refine the hierarchical diffusion framework by further exploring how to more effectively represent and integrate subgoal constraints as conditional information. We also plan to extend the SIHD framework to other offline RL environments and broader diffusion-based generative modeling domains.

## Acknowledgments

The corresponding author is Yicheng Pan. This work is partly supported by National Key R&D Program of China through grant 2021YFB3500700, NSFC through grants 62322202, 62441612 and 62432006, Local Science and Technology Development Fund of Hebei Province Guided by the Central Government of China through grants 246Z0102G and 254Z9902G, the Pioneer and Leading Goose R&D Program of Zhejiang through grant 2025C02044, National Key Laboratory under grant 241-HF-D07-01, Beijing Natural Science Foundation through grant L253021, Hebei Natural Science Foundation through grant F2024210008, and Major Science, Technology Special Projects of Yunnan Province through grants 202502AD080012 and 202502AD080006, CCF-DiDi GAIA Collaborative Research Fund through grant 202527, and partly supported by State Key Laboratory of Complex & Critical Software Environment through grant CCSE-2024ZX-20.

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

# A Framework Details

## A.1 Primary Notations

Table 3: Glossary of Notations.

| Notation | Description |
|---|---|
| $\mathcal{M}$ | Markov Decision Process |
| $\mathcal{S}; \mathcal{A}$ | State space; Action space |
| $\mathcal{P}; \mathcal{R}$ | Transition function; Reward function |
| $s; a; r$ | Single state, action, and reward |
| $\mathcal{Q}; \pi$ | Value network; Policy Network |
| $\tau; \mathcal{D}$ | Single trajectory; Offline dataset |
| $\mathcal{N}; \mathcal{I}$ | Gaussian distribution; Diagonal matrix |
| $\epsilon; \beta$ | Gaussian noise; Variance schedule |
| $\mu; \Sigma$ | Mean vector; Covariance matrix |
| $q; p$ | Prior distribution; Posterior distribution |
| $y; g$ | Conditional information; Single sugboal |
| $\mathcal{G}; \mathcal{T}$ | Topology graph; Encoding tree |
| $\mathcal{V}; \mathcal{E}$ | Vertex set; Edge set |
| $\mathcal{H}^{\mathcal{K}}; \mathcal{H}^{\mathcal{T}}; \mathcal{H}$ | Structural entropy; Shannon entropy |
| $\lambda; \nu$ | Root node; Leaf node |
| $\alpha; \mathcal{U}$ | Single node; Node set at specific layer |

## A.2 Tree Optimization.

The HCSE algorithm is executed iteratively through *stretch* and *compress* operations on $\alpha$-triangle subtrees, which are rooted at a node $\alpha$ whose children are leaf nodes, as illustrated in Figure 5(a). In the *stretch* stage, the $\alpha$-triangle subtree is transformed into a binary subtree (Figure 5(b)) by inserting intermediate nodes that pair the children of $\alpha$, ensuring that the binary branching constraint is satisfied. In the *compress* stage, the resulting binary subtree is compressed into a subtree with exactly three layers of nodes (Figure 5(c)), thereby increasing the height of the subtree by one level.

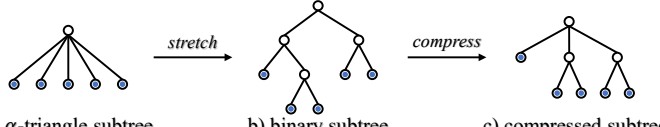

a) $\alpha$-triangle subtree      b) binary subtree     c) compressed subtree

Figure 5: The *stretch* and *compress* optimization on each $\alpha$-triangle subtree.

Algorithm 2 summarizes the optimization procedure applied to each graph $\mathcal{G}$. It begins by initializing a one-layer encoding tree $\mathcal{T}$ for $\mathcal{G}$, and then iteratively selects the set of $h$-layer nodes $\mathcal{U}_h$ that yield the largest structural entropy reduction $\Delta \mathcal{H}^{\mathcal{T}}(\mathcal{G}; \mathcal{U}_h)$. This reduction is achieved by applying *stretch* and *compress* operations to each $\alpha$-triangle subtree, where $\alpha \in \mathcal{U}_h$. The process continues until the desired tree height $\mathcal{K}$ is reached.

---

**Algorithm 2** The HCSE Optimization

---

1: **Input:** a graph $\mathcal{G}, \mathcal{K} \in \mathbb{Z}^+$
2: **Output:** the optimal encoding tree $\mathcal{T}^*$
3: Initialize the one-layer encoding tree $\mathcal{T}$ for $\mathcal{G}$
4: **while** $h_{\mathcal{T}} \leq \mathcal{K}$ **do**
5:     Select the $\mathcal{U}_h$ that maximizes $\Delta \mathcal{H}^{\mathcal{T}}(\mathcal{G}; \mathcal{U}_h)$
6:     **if** $\Delta \mathcal{H}^{\mathcal{T}}(\mathcal{G}; \mathcal{U}_h) = 0$ **then**
7:        Break
8:     **end if**
9:     **for** $\alpha \in \mathcal{U}_h$ **do**
10:        Execute *stretch* and *compress* optimization on the $\alpha$-triangle subtree
11:     **end for**
12: **end while**
13: Return the resulting encoding tree $\mathcal{T}$ as $\mathcal{T}^*$

---

### A.3 Probability Calculation

To compute the joint probability $p_{\theta_1}(s_t, s_{t+1})$ for any adjacent pair of states, we proceed as follows. We begin by sampling initial Gaussian noise trajectories $\tau_{g,K}^{1,1} \sim \mathcal{N}(0, \mathcal{I})$. We then apply the reverse process described in Equation 11 to iteratively denoise these trajectories, yielding $n$ denoised trajectories $\tau_{g,0}^{1,i}\big|_{i=1}^{n}$. From each denoised trajectory, we extract the states at time steps $t$ and $t+1$, constructing a sample set $(s_t^i, s_{t+1}^i)_{i=1}^{n}$. Finally, we apply a two-dimensional Gaussian kernel density estimator to approximate the joint distribution $p_{\theta_1}(s_t, s_{t+1})$.

### A.4 Trajectory Segmentation

---

**Algorithm 3** Hierarchical Trajectory Segmentation

---

1: **Input:** the $\mathcal{K}$-layer encoding tree $\mathcal{T}_s^*$, each historical trajectory $\tau_0$
2: **Output:** hierarchical trajectory segments $\{\tau_{sa}^{h,i}\}_{h=1}^{\mathcal{K}}$ with $1 \le i \le l_g^h$
3: **for** $h \leftarrow \mathcal{K}$ to $1$ **do**
4:     Initialize the first segment $\tau_{sa}^{h,1} \leftarrow [[s_1], [a_1]]$ and the index variable $i \leftarrow 1$
5:     Extract the node set $\mathcal{U}_h$ at the $h$-th layer
6:     **for** $t \leftarrow 2$ to $T$ **do**
7:         **for** $\alpha \in \mathcal{U}_h$ **do**
8:             **if** $s_{t-1} \in \mathcal{V}_\alpha$ and $s_t \in \mathcal{V}_\alpha$ **then**
9:                 $\tau_{sa}^{h,i}[0, :] \leftarrow \tau_{sa}^{h,i}[0, :] + [s_t]$
10:                $\tau_{sa}^{h,i}[1, :] \leftarrow \tau_{sa}^{h,i}[1, :] + [a_t]$
11:             Break
12:             **else if** $s_{t-1} \in \mathcal{V}_\alpha$ or $s_t \in \mathcal{V}_\alpha$ **then**
13:                $i \leftarrow i + 1$
14:                $\tau_{sa}^{h,i} \leftarrow [[s_t], [a_t]]$
15:             Break
16:             **end if**
17:         **end for**
18:     **end for**
19: **end for**
20: Return the hierarchical segments $\{\tau_{sa}^{h,i}\}_{h=1}^{\mathcal{K}}$

---

### A.5 Model Training

---

**Algorithm 4** The SIHD Training Algorithm

---

1: **Input:** the offline dataset $\mathcal{D}_{\pi_b}$, the maximal tree height $\mathcal{K}$
2: **Output:** hierarchical diffusion probabilistic models $\{\epsilon_{\theta_h}\}_{h=1}^{\mathcal{K}}$
3: Construct the topology graph $\mathcal{G}_s$ for offline states in $\mathcal{D}_{\pi_b}$
4: Derive the optimal encoding tree $\mathcal{T}_s^*$ via Algorithm 2
5: **for** each trajectory $\tau_0$ in $\mathcal{D}_{\pi_b}$ **do**
6:     $\{\tau_{sa}^{h,i}\}_{h=1}^{\mathcal{K}} \leftarrow$ hierarchically segment $\tau_0$ via Algorithm 3
7:     **for** $h \leftarrow \mathcal{K}$ to $1$ **do**
8:         $\{\tau_g^h\} \leftarrow$ extract the subgoal sequence from $\{\tau_{sa}^{h,i}\}_{i=1}^{l_g^h}$
9:         **for** $i \leftarrow 1$ to $l_g^h$ **do**
10:             $\tau_g^{h,i} \leftarrow$ extract the lower-layer subgoal sequence for $g_i^h \in \tau_g^h$
11:             $y(\tau_g^{h,i}) \leftarrow$ calculate the conditional information of $\tau_g^{h,i}$
12:             $\hat{\tau}_{g,0}^{h,i} \leftarrow$ estimate the denoised sequence based on $\epsilon_{\theta_h}$
13:             Optimize the $h$-th diffuser $\epsilon_{\theta_h}$ by minimizing the training loss $\mathcal{L}(\theta_h)$ in Equation 15
14:         **end for**
15:     **end for**
16: **end for**

---

---

**Algorithm 5** Subgoal Updating Function $f_{su}$

---

1: **Input:** the layer parameter $h$, hierarchical diffusion probabilistic models $\{\epsilon_{\theta_h}\}_{h=1}^{\mathcal{K}}$, hierarchical subgoal sequences $\{\tau_g^{h,1}\}_{h=1}^{\mathcal{K}}$, the maximal cumulative reward $r_{max}$ in $\mathcal{D}_{\pi_b}$

2: **Output:** the updated subgoal sequences $\tau_g^{h,1}$

3: **if** $h = \mathcal{K}$ **then**

4:  Sampling the starting noised sequence $\hat{\tau}_g^{\mathcal{K},1} \sim \mathcal{N}(0, \mathcal{I})$

5:  Update the noised sequence $\hat{\tau}_g^{\mathcal{K},1}[: l_g^{\mathcal{K},1}] \leftarrow \tau_g^{\mathcal{K},1}$

6:  **for** $k \leftarrow K$ to 1 **do**

7:    $\hat{\epsilon} \leftarrow \epsilon_{\theta_{\mathcal{K}}}(\hat{\tau}_{g,k}^{\mathcal{K},1}, (1-\omega)r_{max} + \omega\emptyset, k)$

8:    $(\mu_{\theta_{\mathcal{K}}}, \Sigma_{\theta_{\mathcal{K}}}) \leftarrow$ extract the mean vector and covariance matrix from $\hat{\epsilon}$

9:    $\hat{\tau}_{g,k-1}^{\mathcal{K},1} \sim \mathcal{N}(\mu_{\theta_{\mathcal{K}}}, \beta_k \Sigma_{\theta_{\mathcal{K}}})$

10:    Return the subgoal sequence $\tau_g^{\mathcal{K},1} \leftarrow \tau_g^{\mathcal{K},1} + [\hat{\tau}_{g,0}^{\mathcal{K},1}[l_g^{\mathcal{K},1}]]$

11:  **end for**

12: **else**

13:  **if** $\tau_g^{h+1,1}[-1]$ is satisfied **then**

14:    $\tau_g^{h+1,1} \leftarrow f_{su}(h+1, \{\epsilon_{\theta_h}\}_{h=1}^{\mathcal{K}}, \{\tau_g^{h,1}\}_{h=1}^{\mathcal{K}}, r_{max})$

15:  **end if**

16:  Sampling the starting noised sequence $\hat{\tau}_g^{h,1} \sim \mathcal{N}(0, \mathcal{I})$

17:  Update the noised sequence $\hat{\tau}_g^{h,1}[: l_g^{h,1}] \leftarrow \tau_g^{h,1}$

18:  **for** $k \leftarrow K$ to 1 **do**

19:    $\alpha \leftarrow$ select the $h$-layer node according to $\tau_g^{h+1,1}[-1]$

20:    Calculate the conditional information $y(\hat{\tau}_{g,k}^{h,1})$ via Equation 12

21:    $\hat{\epsilon} \leftarrow \epsilon_{\theta_h}(\hat{\tau}_{g,k}^{h,1}, (1-\omega)y(\hat{\tau}_{g,k}^{h,1}) + \omega\emptyset, k)$

22:    $(\mu_{\theta_h}, \Sigma_{\theta_h}) \leftarrow$ extract the mean vector and covariance matrix from $\hat{\epsilon}$

23:    $\hat{\tau}_{g,k-1}^{h,1} \sim \mathcal{N}(\mu_{\theta_h}, \beta_k \Sigma_{\theta_h})$

24:  **end for**

25:  Update the subgoal sequence $\tau_g^{h,1} \leftarrow \tau_g^{h,1} + [\hat{\tau}_{g,0}^{h,1}[l_g^{h,1}]]$

26: **end if**

---

### A.6 Toy Example

To intuitively illustrate the hierarchical partitioning process, we present a toy example in which SIHD partitions a trajectory of $1000$ timesteps into sub-trajectories at different hierarchy levels $h$. In each sub-trajectory, the final state is treated as that segment's subgoal.

- Level $h = \mathcal{K}$ (top, coarsest): $\{1, 2, \ldots, 1000\}$.
- Level $h = \mathcal{K} - 1$: $\{1, 2, \ldots, 90\}, \{91, 92, \ldots, 200\}, \ldots, \{901, 902, \ldots, 1000\}$.
- Level $h = 2$ (fine): $\{1, 2, \ldots, 30\}, \{31, 32, \ldots, 47\}, \ldots, \{968, 969, \ldots, 1000\}$.
- Level $h = 1$ (finest): $\{1, 2, \ldots, 8\}, \{9, 10, \ldots, 15\}, \ldots, \{995, 996, \ldots, 1000\}$.

### A.7 Implementation Details

Our SIHD framework is built upon the officially released Diffuser codebase[2], leveraging a hierarchical architecture in which each diffuser processes trajectory segments corresponding to state communities from the optimal encoding tree. To ensure consistent training across variable-length sequences, we pad subgoal and state-action segments to fixed sequence lengths—8 for Gym-MuJoCo tasks and 16 for long-horizon navigation tasks—by repeating terminal states. The diffusion backbone employs a Temporal U-Net [Ronneberger et al., 2015, Ajay et al., 2022] with a Gaussian diffusion process, configured with multiscale temporal convolutions of 32 dimensions. Optimization is performed using the Adam optimizer with exponential moving average (EMA) decay of model weights for stable updates. During training, diffusion models are trained with a batch size of 32, while planning phases use a larger batch size of $64$ to improve sample diversity during guided rollouts.

Horizon settings follow established benchmarks [Janner et al., 2022]: $H = 32$ for Gym-MuJoCo; $H \in \{120, 255, 390\}$ for Maze2D (scaling with task complexity); and $H \in \{225, 255, 450\}$ for AntMaze environments. Classifier-free guidance is applied uniformly across tasks with a fixed weight of $0.1$. The regularization coefficient $\eta$ is selected from $0.01, 0.05, 0.1, 0.15, 0.2$ based on empirical analysis (see Appendix C). This configuration ensures adaptability across both short-horizon locomotion and long-horizon navigation tasks while maintaining computational efficiency and stable performance.

### A.8 Limitations

This work proposes the first multi-layer diffusion framework with adaptive temporal scales, leveraging structural information embedded in historical trajectories. This design addresses the strict structure and overreliance on offline datasets observed in existing hierarchical diffusion methods. On large-scale offline datasets, structural entropy-guided state community partitioning introduces significant computational overhead. To mitigate this, we precompute the optimal encoding tree for each dataset and store the layer-wise community partitions in a dictionary-based data structure, avoiding redundant computation and training-time overhead. Subgoal constraints in the diffusion hierarchy are implemented using a straightforward ending-state replacement strategy. We leave more comprehensive exploration of subgoal conditioning strategies to future work. Another important future direction is extending SIHD to more complex offline RL tasks and broader diffusion-based generative modeling domains.

---

[2]`https://github.com/jannerm/diffuser`

# B Theorem Proofs

## B.1 Proof of Theorem 4.1

*Proof.* We begin by recalling the offline RL objective of modeling the posterior distribution over full trajectories $\tau_0$, conditioned on the optimality $\mathcal{O}_{1:l_g^{\mathcal{K}-1}} = 1$, captured via $y(\tau_0)$:

$$p(\tau_0|y(\tau_0)) = p(\tau_0 \mid \mathcal{O}_{1:l_g^{\mathcal{K}-1}} = 1) \propto p(\tau_0) \cdot p(\mathcal{O}_{1:l_g^{\mathcal{K}-1}} = 1 \mid \tau_0). \tag{16}$$

Using the decomposition of $\tau_0$ into subgoal-conditioned segments at layer $\mathcal{K} - 1$, we write:

$$p(\tau_0) = p(s_0) \cdot \prod_{i=1}^{l_g^{\mathcal{K}-1}} p(g_i^{\mathcal{K}-1}) \cdot p(\tau_{sa}^{\mathcal{K}-1,i} \mid g_i^{\mathcal{K}-1}), \tag{17}$$

where each sub-trajectory $\tau_{sa}^{\mathcal{K}-1,i}$ denotes the segment of $\tau_0$ aimed at achieving the subgoal $g_i^{\mathcal{K}-1}$.

Leveraging the posterior likelihood of optimal variable $\mathcal{O}_i$ from Equation 8, we have:

$$p(\tau_0|y(\tau_0)) \propto p(s_0) \cdot \prod_{i=1}^{l_g^{\mathcal{K}-1}} p(g_i^{\mathcal{K}-1}) \exp\left( \sum_{k=1}^{l_{sa}^{\mathcal{K}-1,i}} \mathcal{R}(s_{\sum_{j=1}^{i-1} l_{sa}^{\mathcal{K}-1,j}+k}, a_{\sum_{j=1}^{i-1} l_{sa}^{\mathcal{K}-1,j}+k}) \right) p(\tau_{sa}^{\mathcal{K}-1,i} \mid g_i^{\mathcal{K}-1})$$

$$= \exp\left( \sum_{t=0}^{T} \mathcal{R}(s_t, a_t) \right) \cdot p(s_0) \cdot \prod_{i=1}^{l_g^{\mathcal{K}-1}} p(g_i^{\mathcal{K}-1}) \cdot p(\tau_{sa}^{\mathcal{K}-1,i} \mid g_i^{\mathcal{K}-1}). \tag{18}$$

We define the cumulative reward term for the top-level hierarchy as:

$$y(\tau_g^{\mathcal{K},1}) := \exp\left( \sum_{t=0}^{T} \mathcal{R}(s_t, a_t) \right). \tag{19}$$

Thus:

$$p(\tau_0|y(\tau_0)) \propto p(\tau_g^{\mathcal{K},1}) \cdot y(\tau_g^{\mathcal{K},1}) \cdot \prod_{i=1}^{l_g^{\mathcal{K}-1}} p(\tau_{sa}^{\mathcal{K}-1,i} \mid g_i^{\mathcal{K}-1}). \tag{20}$$

Each term $p(\tau_{sa}^{\mathcal{K}-1,i} \mid g_i^{\mathcal{K}-1})$ is further recursively decomposed into lower-level subgoal trajectories:

$$p(\tau_{sa}^{\mathcal{K}-1,i} \mid g_i^{\mathcal{K}-1}) = p(\tau_g^{\mathcal{K}-1,i}|g_i^{\mathcal{K}-1}) \prod_{k=1}^{l_g^{\mathcal{K}-1,i}} p(\tau_{sa}^{\mathcal{K}-2,\sum_{j=1}^{i-1} l_g^{\mathcal{K}-1,j}+k} | g_{\sum_{j=1}^{i-1} l_g^{\mathcal{K}-1,j}+k}^{\mathcal{K}-2}). \tag{21}$$

Here, $y(\tau_g^{\mathcal{K}-1,i})$ enforces the consistency of the lower-level subgoals with the higher-level goal $g_i^{\mathcal{K}-1}$ using a Dirac delta function:

$$y(\tau_g^{\mathcal{K}-1,i}) = \delta_{g_i^{\mathcal{K}-1}}(g_{\sum_{j=1}^{i-1} l_g^{\mathcal{K}-1,j}+1}^{\mathcal{K}-2}, \ldots, g_{\sum_{j=1}^{i} l_g^{\mathcal{K}-1,j}}^{\mathcal{K}-2}) = \begin{cases} +\infty & \text{if } g_{\sum_{j=1}^{i} l_g^{\mathcal{K}-1,j}}^{\mathcal{K}-2} = g_i^{\mathcal{K}-1}, \\ 0 & \text{otherwise.} \end{cases} \tag{22}$$

Thus we can write:

$$p(\tau_g^{\mathcal{K}-1,i}|g_i^{\mathcal{K}-1}) = p(\tau_g^{\mathcal{K}-1,i})y(\tau_g^{\mathcal{K}-1,i}). \tag{23}$$

Plugging this into our earlier expansion:

$$p(\tau_0|y(\tau_0)) \propto p(\tau_g^{\mathcal{K},1}) \cdot y(\tau_g^{\mathcal{K},1}) \cdot \prod_{i=1}^{l_g^{\mathcal{K}-1}} p(\tau_g^{\mathcal{K}-1,i})y(\tau_g^{\mathcal{K}-1,i}) \cdot \prod_{i=1}^{l_g^{\mathcal{K}-2}} p(\tau_{sa}^{\mathcal{K}-2,i}|g_i^{\mathcal{K}-2}). \tag{24}$$

By recursively applying this decomposition across all levels $h = H$ down to $h = 1$, we obtain:

$$p(\tau_0|y(\tau_0)) \propto p(\tau_g^{\mathcal{K},1})y(\tau_g^{\mathcal{K},1}) \cdot \prod_{h=1}^{\mathcal{K}-1} \prod_{i=1}^{l_g^h} p(\tau_g^{h,i})y(\tau_g^{h,i}). \tag{25}$$

$\square$

## B.2 Proof of Theorem 4.2

*Proof.* Let $\alpha$ be an intermediate node (i.e., non-root and non-leaf) in the encoding tree $\mathcal{T}_s^*$, and let $\{\alpha_i\}_{i=1}^{l_\alpha}$ denote its children. The structural entropy contribution of $\alpha$ and its children is given by:

$$\mathcal{H}^{\mathcal{T}_s^*}(\mathcal{G}_s';\alpha) + \sum_{i=1}^{l_\alpha} \mathcal{H}^{\mathcal{T}_s^*}(\mathcal{G}_s';\alpha_i) = -\frac{g_\alpha}{\text{vol}(\lambda)} \log \frac{\text{vol}(\alpha)}{\text{vol}(\alpha^-)} - \sum_{i=1}^{l_\alpha} \frac{g_{\alpha_i}}{\text{vol}(\lambda)} \log \frac{\text{vol}(\alpha_i)}{\text{vol}(\alpha)}$$

$$= -\frac{g_\alpha - \sum_{i=1}^{l_\alpha} g_{\alpha_i}}{\text{vol}(\lambda)} \log \frac{\text{vol}(\alpha)}{\text{vol}(\alpha^-)} - \sum_{i=1}^{l_\alpha} \frac{g_{\alpha_i}}{\text{vol}(\lambda)} \log \frac{\text{vol}(\alpha_i)}{\text{vol}(\alpha^-)}.$$
(26)

The total structural entropy of $\mathcal{G}_s'$ under $\mathcal{T}_s^*$ can be expressed as:

$$\mathcal{H}^{\mathcal{T}_s^*}(\mathcal{G}_s') = \sum_{\alpha \in \mathcal{T}_s^*, \alpha \neq \lambda} \mathcal{H}^{\mathcal{T}_s^*}(\mathcal{G}_s';\alpha) = \sum_{h=0}^{\mathcal{K}-1} \sum_{\alpha \in \mathcal{U}_h} \mathcal{H}^{\mathcal{T}_s^*}(\mathcal{G}_s';\alpha)$$

$$= \sum_{\alpha \in \mathcal{U}_{\mathcal{K}-1}} \left[ \mathcal{H}^{\mathcal{T}_s^*}(\mathcal{G}_s';\alpha) + \sum_{i=1}^{l_\alpha} \mathcal{H}^{\mathcal{T}_s^*}(\mathcal{G}_s';\alpha_i) \right] + \sum_{h=0}^{\mathcal{K}-3} \sum_{\alpha \in \mathcal{U}_h} \mathcal{H}^{\mathcal{T}_s^*}(\mathcal{G}_s';\alpha)$$

$$= -\sum_{\alpha \in \mathcal{U}_{\mathcal{K}-1}} \left[ \frac{g_\alpha - \sum_{i=1}^{l_\alpha} g_{\alpha_i}}{\text{vol}(\lambda)} \log \frac{\text{vol}(\alpha)}{\text{vol}(\lambda)} + \sum_{i=1}^{l_\alpha} \frac{g_{\alpha_i}}{\text{vol}(\lambda)} \log \frac{\text{vol}(\alpha_i)}{\text{vol}(\lambda)} \right] + \sum_{h=0}^{\mathcal{K}-3} \sum_{\alpha \in \mathcal{U}_h} \mathcal{H}^{\mathcal{T}_s^*}(\mathcal{G}_s';\alpha)$$

$$= -\sum_{h=1}^{\mathcal{K}-1} \sum_{\alpha \in \mathcal{U}_h} \frac{g_\alpha - \sum_{i=1}^{l_\alpha} g_{\alpha_i}}{\text{vol}(\lambda)} \log \frac{\text{vol}(\alpha)}{\text{vol}(\lambda)} + \sum_{\alpha \in \mathcal{U}_0} \frac{g_\alpha}{\text{vol}(\lambda)} \cdot \log \frac{\text{vol}(\alpha)}{\text{vol}(\lambda)}$$

$$= \sum_{s \in S} p_{\theta_1}(s) \log p_{\theta_1}(s) - \sum_{h=1}^{\mathcal{K}-1} \sum_{\alpha \in \mathcal{U}_h} \frac{g_\alpha - \sum_{i=1}^{l_\alpha} g_{\alpha_i}}{\text{vol}(\lambda)} \log \frac{\text{vol}(\alpha)}{\text{vol}(\lambda)}$$

$$= \mathcal{H}(S) - \sum_{h=1}^{\mathcal{K}-1} \sum_{\alpha \in \mathcal{U}_h} \frac{g_\alpha - \sum_{i=1}^{l_\alpha} g_{\alpha_i}}{\text{vol}(\lambda)} \log \frac{\text{vol}(\alpha)}{\text{vol}(\lambda)}.$$
(27)

Since for each intermediate node $\alpha$, the condition $g_\alpha - \sum_{i=1}^{l_\alpha} g_{\alpha_i} \leq 0$ holds, the upper bound follows:

$$\mathcal{H}^{\mathcal{T}_s^*}(\mathcal{G}_s') \leq \mathcal{H}(S). \tag{28}$$

The structural entropy can further be rewritten in terms of contributions from each layer:

$$\mathcal{H}^{\mathcal{T}_s^*}(\mathcal{G}_s') = \mathcal{H}(S) - \sum_{h=1}^{\mathcal{K}-1} \sum_{\alpha \in \mathcal{U}_h} \frac{g_\alpha - \sum_{i=1}^{l_\alpha} g_{\alpha_i}}{\text{vol}(\lambda)} \log \frac{\text{vol}(\alpha)}{\text{vol}(\lambda)}$$

$$= \mathcal{H}(S) + \sum_{h=1}^{\mathcal{K}-1} \sum_{\alpha \in \mathcal{U}_h} \left[ \frac{\sum_{i=1}^{l_\alpha} g_{\alpha_i} - g_\alpha}{\text{vol}(\alpha)} \cdot \frac{\text{vol}(\alpha)}{\text{vol}(\lambda)} \log \frac{\text{vol}(\alpha)}{\text{vol}(\lambda)} \right]$$

$$\geq \mathcal{H}(S) + \sum_{h=1}^{\mathcal{K}-1} \left[ \max_{\alpha \in \mathcal{U}_h} \frac{\sum_{i=1}^{l_\alpha} g_{\alpha_i} - g_\alpha}{\text{vol}(\alpha)} \cdot \sum_{\alpha \in \mathcal{U}_h} \frac{\text{vol}(\alpha)}{\text{vol}(\lambda)} \log \frac{\text{vol}(\alpha)}{\text{vol}(\lambda)} \right] \tag{29}$$

$$= \mathcal{H}(S) - \sum_{h=1}^{\mathcal{K}-1} \left[ \max_{\alpha \in \mathcal{U}_h} \frac{\sum_{i=1}^{l_\alpha} g_{\alpha_i} - g_\alpha}{\text{vol}(\alpha)} \cdot \mathcal{H}(\mathcal{U}_h) \right]$$

$$= \mathcal{H}(S) - \sum_{h=1}^{\mathcal{K}-1} \left[ \eta_h \cdot \mathcal{H}(\mathcal{U}_h) \right].$$

Therefore, the following lower bound holds:

$$\mathcal{H}^{\mathcal{T}_s^*}(\mathcal{G}_s') \geq \mathcal{H}(S) - \sum_{h=1}^{\mathcal{K}-1} \left[ \eta_h \cdot \mathcal{H}(\mathcal{U}_h) \right]. \tag{30}$$

$\square$

## C    Additional Experiments

### C.1    Sensitivity Analysis

To investigate the effect of key parameters, such as the regularization weight $\eta$ and maximum tree height $\mathcal{K}$, on SIHD performance, we conduct sensitivity analysis experiments on the HalfCheetah and AntMaze tasks. As shown in Figure 6, the decision-making performance of SIHD initially improves as parameter values increase, but plateaus beyond a certain threshold and may decline with further increments. In the HalfCheetah task, smaller values of $\eta$ for the Expert dataset yield better average rewards, as the offline trajectories are generated by a well-trained behavioral policy, requiring minimal additional regularization. Conversely, for the Medium dataset, larger values of $\eta$ lead to better performance due to the lower quality of the behavioral policy. In the AntMaze task, longer planning horizons increase the decision-making challenge, and using more diffusion hierarchy layers results in higher average rewards. However, excessively many diffusion layers shorten the length of individual subgoal sequences, degrading the quality of sequence modeling. Based on these findings, we selected $\mathcal{K} = 4$ as a balanced and effective parameter setting.

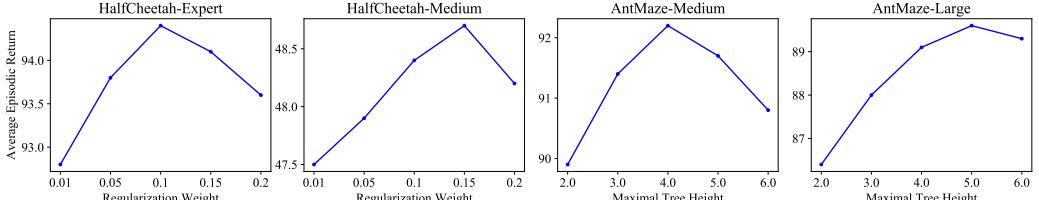

Figure 6: Sensitivity analysis of the regularization weight $\eta$ and maximum tree height $K$ on the Expert and Medium datasets of the HalfCheetah task, and the Medium and Large datasets of the AntMaze task.

### C.2    Computational Efficiency

To evaluate computational efficiency and practicality, we compare the training and planning time of SIHD with baseline methods, including Diffuser and HD, across different Maze2D datasets. To ensure a fair comparison, we use the publicly available source code and hyperparameter configurations, and conduct all experiments on an Nvidia A800 GPU. The experimental results are reported in Table 4. Despite the addition of additional diffusion hierarchy layers, SIHD maintains training and planning efficiency comparable to the 2-layer hierarchical diffusion (HD) baseline. Compared with the flat Diffuser baseline, SIHD achieves $82.5\%$ and $54.8\%$ reductions in training and planning time, respectively, while preserving the computational efficiency benefits of hierarchical diffusion.

Table 4: Comparison of wall-clock time between SIHD and baseline methods (Diffuser and HD) on the U-Maze, Medium, and Large datasets of the Maze2D task.

| Methods | Training (s) | | | Planning (s) | | |
|---|---|---|---|---|---|---|
| | U-Maze | Medium | Large | U-Maze | Medium | Large |
| Diffuser | 11.7 | 48.3 | 42.2 | 0.8 | 4.7 | 4.9 |
| HD | 5.4 | 5.8 | 6.1 | 0.3 | 1.9 | 2.5 |
| SIHD | 5.9 | 6.0 | 6.0 | 0.5 | 1.6 | 2.6 |

To further examine the effect of planning horizon on computational cost, we conduct additional experiments on the Medium-Expert Hopper task with horizon settings of $32$, $48$, and $64$, as summarized in Table 5. While SIHD incurs a slightly higher computational overhead than the HD baseline in most cases, this additional cost remains modest and does not scale with longer horizons, demonstrating the practicality and scalability of the approach.

Table 5: Comparison of wall-clock time between SIHD and baseline methods (Diffuser and HD) on the Expert, Medium, and Replay datasets of the Hopper task under varying planning horizons.

| Methods | Training (s) | | | Planning (s) | | |
|---|---|---|---|---|---|---|
| | $H = 32$ | $H = 48$ | $H = 64$ | $H = 32$ | $H = 48$ | $H = 64$ |
| Diffuser | 6.1 | 8.9 | 11.2 | 0.8 | 1.2 | 1.5 |
| HD | 4.4 | 5.2 | 5.9 | 0.5 | 0.7 | 0.8 |
| SIHD | 4.6 | 5.5 | 6.0 | 0.6 | 0.7 | 0.9 |

### C.3 Ablation Study

Indeed, additional diffusion layers introduce more capacity or parameters, which may influence the performance advantages of SIHD. To address concerns about model capacity, we perform an additional ablation study on the single-task Maze2D benchmark, in which we reduce the parameter count of each diffusion layer so that the total number of parameters across all layers matches that of the two-layer HD model. As shown in Table 6, even without increasing overall model capacity, increasing the number of hierarchical layers yields consistent, significant improvements in performance. This demonstrates that the benefit of SIHD arises from its hierarchical structure rather than simply from an increase in parameter count.

Table 6: Performance comparison between SIHD and baseline methods (HDMI and HD) with similar parameter counts on the Maze2D-U, Maze2D-Medium, and Maze2D-Large datasets.

| Methods | U-Maze | Medium | Large |
|---|---|---|---|
| HDMI | $120.1 \pm 2.5$ | $121.8 \pm 1.6$ | $128.6 \pm 2.9$ |
| HD | $128.4 \pm 3.6$ | $135.6 \pm 3.0$ | $155.8 \pm 2.5$ |
| SIHD | $140.7 \pm 2.1$ | $142.5 \pm 2.9$ | $157.3 \pm 2.0$ |

To assess the impact of community quality on performance, we introduce an ablation variant, SIHD-FT, which replaces structural entropy-based partitioning with a fixed-interval temporal partitioning strategy. As shown in Table 7, SIHD-FT yields a clear drop in average return compared to the full SIHD model. This demonstrates that minimizing structural entropy improves partitioning accuracy and plays a critical role in identifying key offline states and guiding the construction of effective hierarchical diffusion hierarchies.

Table 7: Performance comparison between SIHD and the ablation variant, SIHD-FT on the Maze2D-U, Maze2D-Medium, and Maze2D-Large datasets.

| Methods | U-Maze | Medium | Large |
|---|---|---|---|
| SIHD-FT | $149.17 \pm 1.7$ | $146.8 \pm 2.1$ | $167.2 \pm 4.8$ |
| SIHD | $157.0 \pm 0.6$ | $156.8 \pm 1.7$ | $169.4 \pm 2.7$ |

## C.4 Qualitative Comparison

First, we qualitatively analyze the subgoal sequences sampled by SIHD and the HDMI baseline in a Maze2D navigation task, which requires navigating from a green starting point to a red goal point. For visualization purposes, we display the lowest-level subgoals extracted by SIHD. As shown in Figure 7, the subgoals generated by the single time-scale diffusion hierarchy in HDMI are largely fixed and may not correspond to task-relevant states. In contrast, the multi-scale trajectory segmentation and adaptive subgoal extraction in SIHD effectively identify key states critical for task completion, such as turning points along the navigation path, leading to superior decision-making performance.

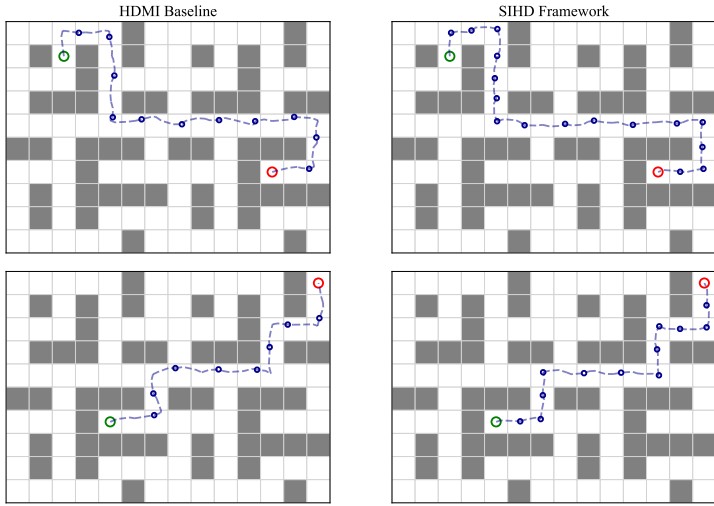

Figure 7: Visualization of sampled subgoals comparing SIHD and the HDMI baseline in the Maze2D navigation task from the starting state (green color) to the goal state (red color).

Second, we visualize the offline trajectories in the Maze2D navigation task, together with trajectories sampled from SIHD and two classical baselines, Diffuser and HD. As shown in Figure 8, the offline dataset includes diverse but suboptimal paths with substantial repeated and invalid state explorations. The trajectory generated by Diffuser does not fully cover the offline dataset and displays relatively simple behavior patterns. The HD baseline achieves full coverage; however, repeated and invalid explorations persist and may even intensify due to overreliance on the offline dataset. In comparison, SIHD generates diverse trajectories while substantially reducing repeated exploration of invalid states. This improvement is attributed to the structural entropy regularizer, which mitigates overreliance on the offline dataset and enhances long-horizon decision-making performance.

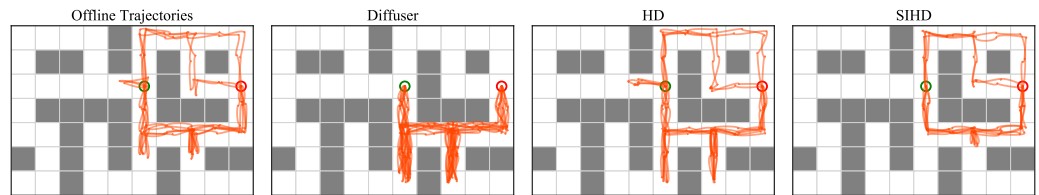

Figure 8: Trajectory visualizations comparing SIHD with classical baselines, Diffuser and HD, in the Maze2D navigation task from the starting state (green color) to the goal state (red color).

