# OpenReview forum: "Structural Information-based Hierarchical Diffusion for Offline Reinforcement Learning"
_NeurIPS.cc/2025/Conference — NeurIPS 2025 poster_

### Official Review · Reviewer_fnUf · 2025-06-23

**Clarity:** 2
**Significance:** 3
**Originality:** 2
**Rating:** 4
**Confidence:** 4

**Summary:**

The authors investigate the classification of states in trajectories from the perspective of the undirected graph, and propose a multi-scale diffusion hierarchy model that simultaneously introduces a structural entropy regularizer to encourage exploration of underrepresented states while avoiding extrapolation errors caused by distributional shifts.

**Questions:**

1) The authors did not explain the meaning of $\alpha^{-}$ in Section 3.3.
2) In line 182, could the authors clarify how k is determined? Additionally, in line 184, since the HCSE optimization algorithm can be used to determine the optimal $\mathcal{K}$, it would be helpful to understand why $\mathcal{K}$ is directly specified in the experimental section.
3) From equation (6), it seems that this partitioning will result in state-action sequences of varying lengths within the trajectory. When h = 1, could the authors clarify whether the corresponding $\tau$ refers to a state-action pair or a sequence of state-action pairs? Additionally, since sequences of fixed length are required for training the diffusion model, how do the authors handle the issue of varying sequence lengths across different layers in the hierarchy?
4) A simple example is helpful to help readers understand this paper:

Suppose that a trajectory: [1, ..., 1000]

h=1: [1, 2, 3, ..., 1000]

h=2: [1,2], [3, 4, 5], [6, 7, 8, 9, 10], ...

h=3: [1,2, 3, 4, 5], [6, 7, 8, 9, 10, 11, 12, 13], [14, ...]

....

Then, 1) whether the separation shown above is right? 2) What are the corresponding goals of each layer ?

5) In Figure 3 (c), should k-1 be replaced with h? Additionally, $l_g^{K-1}$ appears in (c) but is not introduced prior to line 197. Would it be possible for the authors to provide clarification? And in line 202, the authors also do not introduce the meaning of $l_g^{h,i}$.
6) Based on the authors' description, the paper utilizes classifier-free guidance. Could the authors clarify how the conditions of the diffusion model are pre-specified during inference?
7) Could the authors clarify the meaning of "shared" mentioned in line 226? My understanding is that each layer represents a new diffusion model.
8) In line 230, I suggest important references about class-free guidance, such as

  [1] Classifier-Free Diffusion Guidance.

  [2] More Control for Free! Image Synthesis with Semantic Diffusion Guidance.

9) While I can understand that diffusion models are capable of modeling complex distributions of sequences, I am unclear about how the transition probability is computed. It would be helpful if the authors could provide a more detailed explanation.
10) Would constructing an additional weighted state graph $\mathcal{G}_s^\prime$ incur significant time overhead? How is Shannon entropy calculated? I suggest that the authors provide a more detailed explanation to help readers better understand. Why do the authors only apply entropy regularization term on h=1?
11) I notice that the paper does not provide a table of important hyperparameters. I suggest that the authors include the hyperparameter settings related to the experiments and methods.
12) Could the authors clarify how much computational resource is required for state community partitioning on the D4RL dataset?
13) In Table 4, intuitively, increasing the number of layers in the hierarchical diffusion model should directly increase the time for one generation. However, SIHD takes less time than HD. Why is this the case?
14) I noticed that the authors mention online decision-making in Figure 2, but the experimental results do not include online RL. Therefore, I suggest that the authors either add relevant online RL experiments or revise Figure 2.

**Ethical Concerns:**

["NO or VERY MINOR ethics concerns only"]

**Final Justification:**

The authors have addressed my concerns.

**Limitations:**

yes

**Quality:**

3

**Strengths And Weaknesses:**

Pros: The theoretical framework is well-developed, the paper is well-organized, and the experiments are fairly comprehensive.
Cons: However, some parts of the paper are expressed ambiguously, making it difficult for readers to fully understand the intended meaning.

---

> ### Author Rebuttal · Authors · 2025-07-31
>
> Thank you very much for your valuable and insightful comments. They are extremely helpful in improving the quality of our paper. We have systematically and directly addressed each of the Weaknesses (W) and Questions (Q) you raised. Within the constraints of the rebuttal’s word limit and anonymity requirements, we have made every effort to provide thorough and detailed responses to clarify our contributions and resolve your concerns.
>
> $\bullet$ Q1: We have updated Section 3.3 to clarify our structural‑information notation by explicitly defining $\alpha^-$ as the parent of any non‑root tree node $\alpha$.
>
> $\bullet$ Q2: For Line 182, we determine $k$ by sweeping over all plausible values in $[1, |\mathcal{S}|]$, constructing for each state vertex a $k$‑nearest‑neighbor graph (keeping its $k$ most similar neighbors) and computing the resulting one‑dimensional structural entropy. We then select the $k$ that maximizes this entropy. A pseudocode description will be provided in Appendix A of the camera‑ready version after conference acceptance.
>
> As for Line 184, $\mathcal{K}$ denotes the maximum depth of both the coding tree and the diffusion hierarchy. While HCSE optimizes the partition structure under a fixed depth limit, it does not choose that limit itself. Accordingly, we specify $\mathcal{K}$ upfront in our experiments, with detailed justification and a sensitivity analysis presented in Appendix C.1.
>
> $\bullet$ Q3: Indeed, our trajectory segmentation is driven by the automatically discovered state communities rather than fixed time intervals, so the resulting sub‑trajectories naturally vary in length. To train the diffusion model—which requires inputs of uniform length—we pad each shorter subsequence with terminal states until it reaches the prescribed length (see lines 483–485 in Appendix A.6). Furthermore, when $h=1$, each trajectory segment corresponds to an entire sequence of state–action pairs (not just a single pair), as explained in the text.
>
> $\bullet$ Q4: Here’s a toy example of how SIHD hierarchically partitions a trajectory of 1000 timesteps into sub‑trajectories at different levels $h$. In each sub‑trajectory, the final state is treated as that segment’s ``sub‑goal."
>
> Level $h=\mathcal{K}$ (top, coarsest): $[1, 2, \dots, 1000]$.
>
> Level $h=\mathcal{K} - 1$: $[1, 2, \dots, 90], [91, 92, \dots, 200], \dots, [901, 902, \dots, 1000]$.
>
> Level $h=2$ (fine): $[1, 2, \dots, 30], \dots, [78, 79, \dots, 90], \dots, [910, 941, \dots, 950], \dots, [968, 969, \dots, 1000]$.
>
> Level $h=1$ (finest): $[1, 2, \dots, 8], [9, 10, \dots, 15], \dots, [988, 989, \dots,994], [995, 996, \dots, 1000]$.
>
> $\bullet$ Q5: In Figure 3(c), $\mathcal{K}-1$ ndeed represents an arbitrary intermediate layer and can be equivalently replaced with $h$. We have revised Figure 3 accordingly to improve clarity.
>
> Additionally, we have updated the main text prior to line 197 to explicitly introduce the subtrajectory segmentation and subgoal extraction for the second-highest layer, as well as the parameter $\tau_g^{h,i}$, which denotes the subgoal trajectory length. To further address your comment, we now clarify the notation $\tau^\mathcal{K}_g$ and $\tau_g^{h,i}$ in the corresponding context to ensure they are well-defined before use.
>
> We agree that including the trajectory partitioning example you suggested would enhance understanding. We will incorporate it into the appendix in the camera-ready version.
>
> $\bullet$ Q6:
> During inference with the highest-level diffusion model, we employ classifier-free guidance by setting the reward signal—used as conditioning during training—to null. This allows the model to generate a sequence of subgoals without explicit reward input. For conditioning the next-level diffusion model, we select the next subgoal from this sequence and use it to compute the corresponding conditional signal.
>
> Specifically, we locate the node in the optimal encoding tree—constructed from offline trajectories—that is closest to the subgoal state in Euclidean space. Based on this matched node (i.e., the state community), we compute the conditioning signal according to Equation 12. This process is repeated hierarchically for each layer until the full trajectory is generated.
>
> $\bullet$ Q7: Here, ``shared" refers to the fact that, for a given hierarchy level $h$, we train a single diffusion model whose parameters are shared across all sub‑goal sequences at that level. In other words, each layer $h$ indeed has its own diffusion network, but within that layer every sub‑trajectory’s generation uses the same model weights—i.e., the model is shared across sub‑goals at level $h$, rather than having a distinct network for each individual segment.
>
> $\bullet$ Q8: Thank you for the suggestion. We have added citations to Classifier‑Free Diffusion Guidance [1] and More Control for Free! Image Synthesis with Semantic Diffusion Guidance [2] at line 230.
>
> $\bullet$ Q9:  To compute the joint probability $p_{\theta_1}(s_t, s_{t+1})$ between any adjacent state pair, we proceed as follows: we begin by sampling Gaussian noise sequences $\tau_{g, K}^{1,1} \sim \mathcal{N}(0, \mathcal{I})$. These represent initial noisy trajectories. We then apply the reverse process described in Equation 11 to iteratively denoise these trajectories, producing $n$ predicted sequences $\tau_{g, 0}^{1,i}$ with $i=1, \dots, n$. From each denoised sequence, we extract the states at time steps $t$ and $t+1$, forming a sample set $\\{s_t^i, s_{t+1}^i\\}$. Finally, we use a two-dimensional Gaussian kernel density estimator over this sample set to estimate the joint distribution $p_{\theta_1}(s_t, s_{t+1})$.
>
> For clarity and reproducibility, we will include the complete procedure for joint probability estimation as pseudocode in Appendix A, after conference acceptance.
>
> $\bullet$ Q10: As explained previously, constructing the weighted state graph $\mathcal{G}_s^\prime$ requires estimating the joint probability between all adjacent state pairs in the offline trajectories, which is computationally expensive. To mitigate this overhead, we derive in Theorem 4.2 a lower bound of the structural entropy in the form of Shannon entropy. This formulation not only provides theoretical justification for the entropy regularization but also yields an efficient estimation strategy.
>
> Specifically, we leverage the $n$ denoised sequences $\\{\tau_{g, 0}^{1,i}\\}$ obtained from the base diffusion model $\epsilon_{\theta_1}$. The states $s_t^i$ from these sequences are hierarchically clustered according to their Euclidean distances to the layer-wise centroids of the encoding tree $\mathcal{T}_s^*$, yielding community sets $\\{\mathcal{U}_h\\}$ with $h=1,\dots,\mathcal{K}-1$. For each community set $\mathcal{U}_h$, we approximate its Shannon entropy $\mathcal{H}(\mathcal{U}_h)$ using a $k$-nearest neighbors entropy estimator [3], which averages the Euclidean distances between each community and its $k$-th nearest neighbor, as detailed in Equation 15.
>
> Since the entire entropy computation is based solely on the lowest-level diffusion model $\epsilon_{\theta_1}$, the entropy regularization is applied only at $h=1$. This design choice ensures computational efficiency while still enforcing meaningful structure through the regularization.
>
> $\bullet$ Q11: We have added a description of the key hyperparameter settings in Appendix A and will ensure that this information is included in the camera-ready version upon acceptance.
>
> $\bullet$ Q12: All experiments in our work were conducted on five Linux servers, each equipped with an NVIDIA RTX A6000 GPU and an Intel i9-10980XE CPU running at 3.00 GHz. To reduce redundant computation, we performed state community partitioning offline and stored the resulting structures in a dictionary format. This design ensures that the community assignments do not need to be recomputed during training or inference, thereby maintaining the overall computational feasibility of training and deploying the hierarchical diffusion model on the D4RL dataset.
>
> $\bullet$ Q13: Intuitively, increasing the number of layers in a hierarchical diffusion model would introduce additional time overhead. As shown in Tables 4 and 5, SIHD generally requires more training and inference time than the two-layer HD model in most scenarios. However, it is important to note that under a fixed time horizon, adding more hierarchical layers leads to shorter sequence lengths for each individual diffusion model. Furthermore, both training and inference are implemented with parallel computation across layers. As a result, the time overhead introduced by the hierarchical structure is significantly mitigated, and in some cases, SIHD even demonstrates slightly better time efficiency compared to HD.
>
> $\bullet$ Q14: In Figure 2, the term ``online decision-making" refers specifically to the use of the trained hierarchical diffusion model for online inference or trajectory planning, rather than interaction-based online reinforcement learning. To avoid any potential misunderstanding, we have revised Figure 2 accordingly to clarify this intent.
>
> References:
>
> [1] Classifier-Free Diffusion Guidance. NeurIPS 2021 Workshop on Deep Generative Models and Downstream Applications.
>
> [2] More Control for Free! Image Synthesis with Semantic Diffusion Guidance. IEEE/CVF 2023.
>
> [3] Nearest neighbor estimates of entropy. American journal of mathematical and management sciences, 2003.

---

> > ### Comment · Reviewer_fnUf · 2025-08-05
> >
> > I greatly appreciate the authors' detailed response, which has addressed most of my concerns.
> > I believe these clarifications will also help readers understand the paper better. However, some issues still remain unresolved, and I hope the authors can provide further responses.
> >
> > - Q2: Would "sweeping over all plausible values" result in significant time overhead? Additionally, I suggest the authors post the algorithm for k in the rebuttal first.
> > - Q4: In the example of Q4, is it possible that the length division of each layer could be arbitrary, like the example I gave, because in Q3, trajectory segmentation might lead to various lengths.
> > - Q6: In fact, what I want to ask is how structural information gain is determined, because during training, structural information can be calculated from the dataset, but how is it specified during inference?

---

> > > ### Author Response · Authors · 2025-08-05
> > >
> > > Thank you very much for your further feedback. We sincerely appreciate your continued engagement and are more than happy to provide additional clarifications in response to your remaining concerns. We hope the following explanations will fully address your questions.
> > >
> > > $\bullet$ Q2: To clarify, we include here the specific algorithm used for selecting the parameter $k$. The procedure begins with a fully connected graph and iteratively constructs a series of $k$-nearest neighbor graphs by retaining, for each vertex, the $k$ edges with the largest absolute weights. For each candidate $k$, we evaluate the corresponding graph's one-dimensional structure entropy and select the value that globally minimizes this entropy as the final parameter $k^*$ . To address concerns about computational overhead—particularly on large-scale datasets—we impose an upper bound on $k$, set to $20$ in our experiments (i.e., $k \in [1 , 20]$). This constraint effectively limits the time complexity while ensuring that the resulting state graph remains sufficiently sparse for downstream processing. Therefore, the process of ``sweeping over all plausible values" is both tractable and efficient in practice.
> > >
> > > ```pseudocode
> > > input: a weighted, undirected, and complete graph $\mathcal{G}_s$
> > > output: a sparsified graph $\mathcal{G}_s$
> > > for $k = 1$ to $\min {|\mathcal{S}|, 20}$ do
> > >   if the $k$NN graph exists then
> > >       calculate the one-dimensional structural entropy $\mathcal{H}^1(\mathcal{G}_s)$ for $k$
> > >   end if
> > > end for
> > > select the optimal value k by minimizing $\mathcal{H}^1(\mathcal{G}_s)$
> > > ```
> > >
> > > $\bullet$ Q4: In our method, the hierarchical segmentation of offline trajectories is strictly guided by state community partitioning based on structure entropy, and as such, the resulting segment lengths can indeed be arbitrary.
> > >
> > > To ensure that the diffusion model operates on subsequences of uniform length within each layer, we perform a series of preprocessing steps on the segmented trajectories. Specifically, overly long subsequences are divided along the temporal axis into multiple sub-trajectories that meet a predefined length requirement (either 8 or 16), along with a remaining short subsequence. For consecutive short subsequences, if their combined length satisfies the predefined requirement, they are merged into a temporally continuous segment. Finally, any remaining subsequence shorter than the required length is padded with its terminal state until the target length is reached.
> > >
> > > This design ensures that the input to the diffusion model is consistently structured, while preserving the temporal and hierarchical semantics introduced by the initial structural entropy-based segmentation.
> > >
> > > $\bullet$ Q6: During inference, the structural information gain used in the conditional signal is derived from the hierarchical state community partitioning of the offline dataset. For each sub-goal state generated at the higher level, we identify the most similar community at the next lower level by comparing it to the averaged feature representation of each offline community.
> > >
> > > Once the closest matching community is found, we compute its information gain based on the offline-constructed encoding tree. This value is then used as the structural component of the conditional signal, enabling hierarchical reasoning throughout the inference process. This approach ensures consistency between training and inference, while leveraging the offline structural hierarchy to guide goal decomposition effectively.

---

> > > > ### Author Response · Authors · 2025-08-06
> > > >
> > > > Dear Reviewer,
> > > >
> > > > We have provided a thorough and point-by-point response to the concerns you raised. We would greatly value your feedback on whether our rebuttal fully addresses your questions or if any issues remain. Whenever convenient, an early reply would be sincerely appreciated.
> > > >
> > > > Thank you very much for your time and consideration.
> > > >
> > > > Best regards,
> > > >
> > > > Submission7053 Authors

---

> > > > ### Comment · Reviewer_fnUf · 2025-08-08
> > > >
> > > > I appreciate the authors' responses, which have addressed my concerns.
> > > >
> > > > I hope the authors ensure that the clarifications and explanations mentioned above are fully considered and incorporated into the revised version. Good luck.

---

> > > > > ### Author Response · Authors · 2025-08-08
> > > > >
> > > > > Thank you for your confirmation—we’re glad our response resolved your concerns. All clarifications and explanations from the rebuttal will be fully integrated into the revised paper.

---

### Official Review · Reviewer_uo2i · 2025-07-01

**Clarity:** 3
**Significance:** 3
**Originality:** 3
**Rating:** 4
**Confidence:** 2

**Summary:**

In this paper, a Structural Information-based Hierarchical Diffusion framework, namely SIHD is proposed, where structural information embedded in offline trajectories is utilized to construct the diffusion hierarchy. Furthermore, a structural entropy regularizer is presented to encourage exploration of underrepresented states. Extensive experiments were conducted on D4RL to show the effectiveness and generalization of the proposed method.

**Questions:**

Please refer to the Weaknesses part of the review.

**Ethical Concerns:**

["NO or VERY MINOR ethics concerns only"]

**Final Justification:**

The authors' response has addressed some of my concerns, and I'd like to keep the positive score of the paper. But it still seems the proposed framework is limited to some offline datasets that can be partitioned in a tree-structured manner, which may not be prevalent in reality.

**Limitations:**

Yes.

**Paper Formatting Concerns:**

No major formatting issues

**Quality:**

3

**Strengths And Weaknesses:**

Strengths:
The presented idea that using structural information existed in the historical trajectories for offline policy learning is novel and extensive experiments are conducted to show the effectiveness of the proposed framework. Moreover, theoretical guarantees are provided to enable the decomposition of the conditional generation problem for offline trajectories into a hierarchical diffusion process.

Weaknesses:
The main concerns are the scalability and generalization of the proposed framework. Besides D4RL, it is unclear whether the required structure information for the deriving of the tree-structured partitioning of state communities exists in other offline datasets. Also, how can we guarantee the accuracy of the partitioning of the state communities?  Does it have a clear impact on the final performance of the presented method? As is shown in the experiments, the community partitioning is time-consuming and computation-expensive, it may be hardly to be extended to large-scale systems, experiments on some real-world applications may be necessary to verify the method’s scalability. Furthermore, only experimental results are provided to support the performance of the structural entropy regularizer while theoretical guarantees are lacked.

---

> ### Author Rebuttal · Authors · 2025-07-31
>
> Thank you very much for your valuable and insightful comments. They are extremely helpful in improving the quality of our paper. We have systematically and directly addressed each of the Weaknesses (W) and Questions (Q) you raised. Within the constraints of the rebuttal’s word limit and anonymity requirements, we have made every effort to provide thorough and detailed responses to clarify our contributions and resolve your concerns.
>
> $\bullet$ W1: To ensure fair comparisons, our initial experiments focused on benchmarks commonly used in hierarchical diffusion research, most of which are based on D4RL. However, to address the reviewer’s concern regarding generalizability, we conducted additional evaluations on the FrankaKitchen and NeoRL [1] benchmarks.
>
> The results demonstrate that SIHD consistently achieves higher average returns across these diverse offline tasks. This not only reinforces the performance and generality of our framework but also empirically validates the feasibility of deriving hierarchical structures based on structural information from offline states beyond D4RL. These findings suggest that the state-space structure necessary for tree-structured partitioning can be reliably inferred from a variety of offline datasets, highlighting the scalability and broad applicability of our approach.
>
> |FrankaKitchen|Partial|Mixed|
> |:-:|:-:|:-:|
> |Diffuser|$56.2 \pm 5.4$|$50.0 \pm 8.8$|
> |HD|$73.3 \pm 1.4$|$71.7 \pm 2.7$|
> |SIHD|$76.8 \pm 1.7$|$74.0 \pm 2.3$|
>
> |NeoRL|FinRL-L-99|FinRL-L-999|FinRL-M-99|FinRL-M-999|
> |:-:|:-:|:-:|:-:|:-:|
> |MB-PPO|$328$|$656$|$1213$|$698$|
> |HDMI|$415$|$733$|$1007$|$754$|
> |SIHD|$457$|$760$|$1294$|$772$|
>
> $\bullet$ W2: In our method, the accuracy of state community partitioning is ensured through the minimization of structural entropy, which serves as an unsupervised optimization objective for identifying coherent hierarchical communities. This approach has been widely adopted in hierarchical decision-making [2] and intrinsic reward design [3] within reinforcement learning.
>
> To further address your concern, we quantitatively evaluated the quality of the partitioned communities on the Maze2D task using two standard unsupervised clustering metrics: Silhouette score (S) and Calinski-Harabasz index (CH). The results show that our structural entropy-based partitioning achieves $S>0.8$ and $CH>170$, indicating strong intra-community cohesion and inter-community separation—both hallmarks of high-quality clustering.
>
> To assess the impact of community quality on performance, we designed an ablation variant, SIHD-FT, which replaces structural entropy-based partitioning with fixed-interval temporal segmentation. Experimental results show a clear drop in average return for SIHD-FT compared to the full SIHD model. This confirms that minimizing structural entropy not only improves partitioning accuracy but also plays a critical role in identifying key offline states and guiding the construction of effective hierarchical diffusion structures.
>
> |Multi-task Maze2D|U-Maze|Medium|Large|
> |:-:|:-:|:-:|:-:|
> |SIHD-FT|$149.3 \pm 1.7$|$146.8 \pm 2.1$|$167.2 \pm 4.8$|
> |SIHD|$157.0 \pm 0.6$|$156.8 \pm 1.7$|$169.4 \pm 2.7$|
>
> $\bullet$ W3: You are absolutely right that community partitioning introduces additional computational overhead, particularly in large-scale settings. To address this concern, we designed our system to perform state community partitioning entirely offline, storing the resulting structure as a dictionary. This avoids repeated computation during training and inference, ensuring that the hierarchical diffusion model remains practically deployable.
>
> In Appendix C.2 (Computational Efficiency), we present a detailed comparison of training and inference time between SIHD and baselines such as Diffuser and HD across tasks of varying scale. Despite the inclusion of additional hierarchical layers, SIHD maintains comparable runtime efficiency to HD, thanks to parallelism and the reuse of precomputed structures.
>
> $\bullet$ W4: While we provide strong empirical evidence demonstrating the benefits of the structural entropy regularizer, we also recognize the importance of theoretical support.
>
> To that end, in addition to experimental validation, we include a formal theoretical analysis in Theorem 4.2, which establishes a lower bound formulation of Shannon entropy under a structural entropy framework. This result shows that our regularizer promotes exploration in a controlled manner: it encourages trajectory diversity while preserving the hierarchical community structure inferred from the offline data.
>
> Unlike conventional entropy maximization strategies [4] that risk inducing unsafe out-of-distribution actions, our formulation explicitly constrains exploration within topologically meaningful regions of the state space. This provides a theoretical foundation for the regularizer’s ability to balance generalization with robustness, mitigating extrapolation error in offline settings.
>
> References:
>
> [1] NeoRL: A near real-world benchmark for offline reinforcement learning. NeurIPS 2022.
>
> [2] Hierarchical State Abstraction Based on Structural Information Principles. IJCAI 2023
>
> [3] Effective Exploration Based on the Structural Information Principles. NeurIPS 2024.
>
> [4] Entropy-regularized Diffusion Policy with Q-Ensembles for Offline Reinforcement Learning. NeurIPS 2024.

---

### Official Review · Reviewer_mARi · 2025-07-01

**Clarity:** 1
**Significance:** 3
**Originality:** 3
**Rating:** 4
**Confidence:** 3

**Summary:**

The paper introduces SIHD, a novel hierarchical diffusion method for offline RL. The main idea is to build a multi-scale hierarchy by first partitioning the state space, and thus trajectories in the data, into tree-based structures, and then train diffusion models at different levels of this hierarchy to enable long-horizon inference. The method is tested on 3 domains from the D4RL benchmark and compares favorable to both non-hierarchical and hierarchical state-of-the-art baselines.

**Questions:**

Please address limitations above

**Ethical Concerns:**

["NO or VERY MINOR ethics concerns only"]

**Final Justification:**

Reasons why I have increased my score and confidence:
- The authors have addressed all concerns about writing quality and updated the paper accordingly, which I think essential to reach the acceptance bar
- The authors have provided additional experiments that I find quite valuable (especially the one about comparing algorithms at the same capacity)

Reasons not to increase further:
- the main algorithmic limitations (eg scalability / complexity) are still there and seem intrinsic to the method. Experiments are quite small-scale to conclude these are not issues

**Limitations:**

yes

**Quality:**

2

**Strengths And Weaknesses:**

Strengths:
1. The paper considers a relevant and well-motivated problem (long-horizon generalization in offline RL)
2. The paper tackles and important limitation of existing hierarchical methods (i.e., the hierarchy is mostly restricted to 2 layers), which limits their scalability to large horizons
3. The proposed method performs well empirically in comparison to both non-hierarchical and hierarchical state-of-the-art baselines
4. Experiments to ablate the main algorithmic design choices are provided, showing that each of them is indeed relevant
5. Existing literature seems to be properly discussed (though I am not an expert of hierarchical methods for offline RL)

Weaknesses:
1. I found the paper very hard to read, especially Section 4 given the convoluted notation with many sub/superscripts. After reading it multiple times, I only got a high-level understanding of the proposed method, while I am not sure I fully grasp all the details to properly judge its significance. Both Figure 2 and 3 didn't help as they only list the main algorithmic components labeled by hard-to-read notation and without explaining their functionalities. I think a figure that more directly shows how a trajectory may be segmented into different hierarchies, which part goes into which diffuser, and what's the conditional information used for those would greatly help.
2. Also, in was not clear what's the conditional information used at the different levels of the hierarchy. Equation 8 seems to imply that the subgoal g_i is used as conditioning, but then the right-hand side doesn't depend on g_i (unless that dependence is hidden in R?). Equation 9 seems to imply that the conditioning is the exponential of the reward for the last level of the hierarchy, while intermediate layers use subgoal information. Finally, Equation 12 seems to imply that intermediate layers actually use the structural information gain
3. One limitation of the proposed method is that it needs to train one different diffusion model for each level of the hierarchy. While it is true this is in the attempt to overcome the limit of 2 layers in existing methods, this brings its own issues: (1) it may be computationally much more expensive to train and use at test-time, (2) it prevents apple-to-apple comparison with existing methods since the proposed one has more capacity/parameters. There is no discussion about this in the paper
4. The motivation and benefits of the structural entropy-based exploration regularizer are not clear. What does it even mean to "explore" in offline RL? Why would we want to go out of the data distribution, when the majority of offline RL works actually try to stay within data support to prevent extrapolation errors?
5. While the whole approach is motivated by enabling long-horizon predictions, the experiments are still done on relatively short-horizon tasks and with only 3-4 hierarchy layers. There is no scalability analysis (in terms of compute/data/etc) for these variables
6. A more qualitative empirical evaluation showing, eg, what the diffusers learn at different hierarchies is missing
7. It was not clear how the proposed method can be used at test time (eg for planning). I think this is relegated to an appendix, but it seems a very important aspect to have in the main paper

---

> ### Author Rebuttal · Authors · 2025-07-31
>
> Thank you very much for your valuable and insightful comments. They are extremely helpful in improving the quality of our paper. We have systematically and directly addressed each of the Weaknesses (W) and Questions (Q) you raised. Within the constraints of the rebuttal’s word limit and anonymity requirements, we have made every effort to provide thorough and detailed responses to clarify our contributions and resolve your concerns.
>
> $\bullet$ W1: To improve clarity and readability, we have made comprehensive revisions to Section 4 of the paper. Specifically, we added clear explanations for the subscripts and superscripts used in our notation—where superscript $h$ and $i$ indicate the diffusion hierarchy level and temporal ordering within that level, and subscripts $g$ and $sa$ refer to subgoal sequences and state-action subtrajectories, respectively.
>
> In response to your comments on Figures 2 and 3, we redesigned Figure 2 to visually separate the three core modules into distinct subfigures, each illustrating the technical workflow of one stage in a more intuitive manner. Additionally, we expanded Figure 3 to provide a more concrete, visual depiction of the full hierarchical diffusion process—from community partitioning, trajectory segmentation, and subgoal extraction, to conditional signal definition—based on offline trajectories.
>
> $\bullet$ W2: In Equation 8, within the control-as-inference paradigm, the subgoal $g_i^{\mathcal{K} - 1}$ is implicitly defined as the terminal state of the corresponding subtrajectory. Specifically, it is set to the cumulative length of preceding subtrajectories, i.e., $g_i^{\mathcal{K} - 1} = \sum_{j=1}^{i} l_{sa}^{\mathcal{K} - 1,j}$. This dependence is not explicitly shown on the right-hand side of Equation 8 but is encoded through the structure of the generated subtrajectory $\tau_g^{h,i}$.
>
> To generalize this idea, Theorem 4.1 formalizes a unified formulation of hierarchical conditional diffusion, where the conditioning term $y(\tau_g^{h,i})$ represents a flexible signal associated with the subgoal state. In prior approaches, this is often instantiated as the cumulative reward over the subtrajectory. However, such a reward-based signal is often insufficient—especially in long-horizon or sparse-reward settings—due to its limited informativeness and locality.
>
> To address this, in our framework:
>
> At the top layer, where the receptive field covers the full trajectory, we retain cumulative return-based conditioning to reflect global objectives.
>
> At intermediate layers, where the model operates on shorter temporal windows with limited context, we introduce a structure-aware signal based on the expected hitting probability of subgoals via random walks on the offline state graph. This formulation, defined in Equation 12, captures meaningful topological relationships in the latent space and serves as a robust and informative form of guidance for lower-level diffusion.
>
> We believe this hierarchical conditioning strategy is key to the improved performance and flexibility of our model, and we will clarify these connections more explicitly in the final version of the paper.
>
> $\bullet$ W3: We acknowledge that introducing additional hierarchical levels may increase the computational burden during both training and inference. To address this, we included a detailed analysis in the original manuscript (Appendix C.2, Computational Efficiency), where we compare the runtime cost of SIHD with baseline methods such as Diffuser and HD. The results show that while SIHD does incur slightly higher training and inference time compared to HD, the overhead is modest and well-justified by the consistent performance gains. More importantly, SIHD retains a significant efficiency advantage over single-layer Diffuser models, reinforcing the practical feasibility and scalability of our approach.
>
> To further address concerns about model capacity, we conducted an additional ablation on Single-task Maze2D in which we reduced the parameter count of each individual diffusion layer so that the total number of parameters across all layers matches that of the two-layer HD model. As summarized in the following table, even without increasing overall model capacity, increasing the number of hierarchical layers yields consistent and significant performance improvements. This demonstrates that the benefit of SIHD arises from its hierarchical structure rather than simply from added model size. We will include these discussions and results more explicitly in the final version of the paper.
>
> |Single-task Maze2D|U-Maze|Medium|Large|
> |:-:|:-:|:-:|:-:|
> |HDMI|$120.1 \pm 2.5$|$121.8 \pm 1.6$|$128.6 \pm 2.9$|
> |HD|$128.4 \pm 3.6$|$135.6 \pm 3.0$|$155.8 \pm 2.5$|
> |SIHD|$140.7 \pm 2.1$|$142.5 \pm 2.9$|$157.3 \pm 2.0$|
>
> $\bullet$ W4: In offline RL, it is indeed essential to avoid excessive extrapolation beyond the support of the dataset, as this can lead to unreliable value estimates. However, prior works [1, 2] have shown that controlled exploration—particularly toward underrepresented but plausible regions of the state-action space—can improve robustness to value estimation errors and enhance generalization, especially when guided by entropy-based regularization.
>
> The challenge, as you noted, lies in balancing this exploration. Unconstrained deviation from the data distribution may introduce harmful out-of-distribution (OOD) behaviors. To address this, we introduce a structural entropy-based regularizer, which differs from conventional entropy maximization approaches in two key ways:
>
> Structural Awareness: Rather than encouraging exploration uniformly across the entire state space, our method promotes diversity within the hierarchical community structure inferred from the offline data. This ensures that the exploration remains aligned with the intrinsic topological structure of the data, as captured by the encoding tree.
>
> Controlled Diversity: The structural entropy metric formalized in Theorem 4.2 not only promotes generative diversity but also penalizes excessive deviation from the established hierarchical state structure. This provides a principled way to balance exploration and support preservation, effectively mitigating extrapolation risks.
>
> Our ablation results further demonstrate that incorporating this regularizer improves performance compared to both no regularization and standard entropy-based exploration.
>
> $\bullet$ W5: In our experimental evaluation, we use trajectory lengths up to 390 in Maze2D and 450 in AntMaze environments—substantially longer than typical short-horizon tasks—demonstrating SIHD’s capacity for long-range planning. To further investigate scalability, we conduct a detailed analysis in Appendix C.1 (Sensitivity Analysis), where we vary the number of diffusion layers from 2 to 6 across different scales of the AntMaze tasks.
>
> The results indicate that increasing the number of hierarchy levels enables planning over longer time spans. However, we also observe that deeper hierarchies result in much shorter subtrajectory lengths per layer, which can degrade the generative performance of individual diffusion models. This trade-off highlights a key design consideration in balancing depth and subtrajectory complexity.
>
> Moreover, in Appendix C.2 (Computational Efficiency), we compare the training and inference time across tasks of varying scales. Despite introducing additional diffusion layers, SIHD maintains computational efficiency comparable to the two-layer HD baseline, due to effective parallelism and reduced per-layer sequence lengths.
>
> $\bullet$ W6: We have expanded our Qualitative Comparison (Figure 7) to illustrate exactly what each diffusion level learns on the Maze2D navigation task: low‑level diffuser generates fine-grained waypoints that navigate local obstacles, mid‑level diffuser proposes intermediate subgoals that bridge across small clusters of states—e.g., navigating between adjoining corridors, and high‑level diffuser identifies long‑range landmarks or critical “turning points” across the entire maze, allowing for flexible, multi‑scale planning.
>
> By visualizing these subgoals side‑by‑side (and contrasting them with those from the two‑layer HDMI baseline), readers can immediately see how additional hierarchy levels yield increasingly abstract and spatially dispersed goals. Due to rebuttal submission constraints, we cannot include the updated figure here, but it will appear in full in the camera‑ready version after conference acceptance.
>
> $\bullet$ W7: In the current version of the paper, we provide a complete pseudocode implementation of the inference procedure in Appendix A.5. To improve accessibility and clarity, we have now moved a high-level summary of this procedure into the main text and added explanatory commentary to better convey the planning mechanism.
>
> At inference, our SIHD planner proceeds as follows (see Algorithm4): We first initialize each hierarchy’s subgoal buffers and the 1‑layer state–action sequence (lines 3 and 4). Then for each planning step, we (i) sample an initial noisy latent sequence from the diffuser prior (lines 6 and 7), (ii) invoke the subgoal proposer to revise the top‑level goal if its terminal criterion is satisfied (lines 8-10), and (iii) execute a reverse‑diffusion rollout across latent timesteps—at each diffuser step to sample the next latent sequence (lines 11-17). Finally, we decode and integrate the new state and action back into the 1‑layer sequence (lines 18 and 19), and after $H$ iterations output the resulting action trajectory (line 21).
>
> References:
>
> [1] Modeling purposeful adaptive behavior with the principle of maximum causal entropy. Carnegie Mellon University, 2010.
>
> [2] Entropy-regularized Diffusion Policy with Q-Ensembles for Offline Reinforcement Learning. NeurIPS 2024.

---

> > ### Comment · Reviewer_mARi · 2025-08-04
> >
> > Thanks for the detailed response. I have increased my score (see updated review for details)

---

> > > ### Author Response · Authors · 2025-08-05
> > >
> > > Dear Reviewer,
> > >
> > > Thank you for carefully reviewing our rebuttal and for raising your score. We are grateful for the detailed comments you provided in your updated review; they will help us further improve the work.
> > >
> > > Best regards,
> > >
> > > Submission7053 Authors

---

### Official Review · Reviewer_42zk · 2025-07-02

**Clarity:** 3
**Significance:** 4
**Originality:** 3
**Rating:** 4
**Confidence:** 4

**Summary:**

This paper proposes SIHD, a hierarchical diffusion framework for offline reinforcement learning that adaptively builds multi-scale policy hierarchies based on structural information from offline trajectories. By quantifying information gain across state communities and incorporating it into conditional diffusion, SIHD enables flexible and effective long-horizon planning. A structural entropy regularizer further promotes exploration of underrepresented states while mitigating extrapolation errors. Experiments on D4RL benchmarks show that SIHD outperforms prior methods in both decision quality and generalization.

**Questions:**

1. How does the result benefit from the hierarchical diffusion structure compared with simple 2-layer diffusion structure? Can the authors show the difference with experiments like a case study?
2. What is the intuiation of using such a encoding tree to establish the hierarchical partition?

**Ethical Concerns:**

["NO or VERY MINOR ethics concerns only"]

**Final Justification:**

My major concern was mainly related to the writings. Since the authors claim they will improve their writing and they have list the detailed updates, I tend to maintain my positive score.

**Limitations:**

The authors addressed the limitations while they mentioned this work doesn't have any potential negative societal impacts.

**Paper Formatting Concerns:**

There is no major formatting issue.

**Quality:**

4

**Strengths And Weaknesses:**

Strengths:
1. The idea of this paper is novel and solid, improving from the 2-layer structure to the multi-layer structure is promising.
2. The method proposed in this paper is interesting, especially about the threorical part. The partition algorithm is great, while the authors need to clarify the intuition of the partition.
3. The empirical result is very strong, the authors show a impressive improvement compared with previous sota algorithms.

Weakness:
1. The writing of this paper needs to be polished. The idea and intuition of each algorithm component is unclear in the paper. Some of the notations is confusing, e.g. sec 4.1.
2.(minor). The figure 2 is hard to read, it would be better if the authors could change the lines in the figure to make them easier to distinguish.
3. Is the improvement of this work compared with previous work incremental?
4. The experiments are limited to D4RL, the authors could add more experiments on different benchmarks like meta-world.

---

> ### Author Rebuttal · Authors · 2025-07-31
>
> Thank you very much for your valuable and insightful comments. They are extremely helpful in improving the quality of our paper. We have systematically and directly addressed each of the Weaknesses (W) and Questions (Q) you raised. Within the constraints of the rebuttal’s word limit and anonymity requirements, we have made every effort to provide thorough and detailed responses to clarify our contributions and resolve your concerns.
>
> $\bullet$ W1: We have revised the paper to improve the clarity of writing and the presentation of algorithmic components. Specifically, we clarified the notation used throughout the paper: superscripts $h$ and $i$ denote the diffusion layer and temporal index within the same layer, respectively, while subscripts $g$ and $sa$ indicate subgoal sequences and state-action subtrajectories.
>
> In response to the feedback on Figure 2, we have redrawn the figure to improve its readability. The three key modules are now visualized separately in distinct subfigures, allowing for a clearer depiction of the processing flow at each stage. Additionally, we expanded the introductory paragraph of Section 4 to briefly explain the roles of the three main modules:
>
> Hierarchy Construction Module: captures the topological structure of offline states based on feature similarity and automatically uncovers hierarchical communities by optimizing structural entropy. These communities are then used to construct a flexible, multi-scale diffusion hierarchy.
>
> Conditional Diffusion Module: inputs temporally segmented trajectories into corresponding layers of a shared diffusion model, integrating quantized rewards and structural signals to perform forward diffusion and reverse prediction at multiple time scales.
>
> Regularized Exploration Module: leverages structural entropy as a measure of generative diversity and introduces a regularization loss to encourage the exploration of underrepresented offline states during training and inference.
>
> These changes are intended to make the overall pipeline and technical intuition easier to follow.
>
> $\bullet$ W2: As outlined in the Introduction and Related Work sections, although prior studies [1, 2] have proposed hierarchical diffusion models, they are typically restricted to a fixed two-level structure with uniform temporal segmentation. These methods lack the flexibility to adapt across varying time scales and essentially operate as two-layer diffusion models. In contrast, our work addresses a core open challenge in the field—designing a truly general and flexible hierarchical diffusion framework. To the best of our knowledge, this is the first method to realize a multi-level diffusion hierarchy that adapts to diverse temporal abstractions, making our contribution a substantive advancement rather than an incremental improvement.
>
> $\bullet$ W3: To ensure fair comparison with prior work, our initial experimental setup focused on benchmarks commonly adopted in hierarchical diffusion studies [1, 2], the majority of which are based on D4RL. Nevertheless, to address the reviewer’s concern regarding generality, we conducted additional experiments on the FrankaKitchen and NeoRL [3] benchmarks.
>
> In the NeoRL benchmark, we also included MB-PPO [3], a strong model-based baseline known for its performance in this setting. The experimental results demonstrate that SIHD consistently achieves higher average returns across different offline tasks. These findings further substantiate the effectiveness and generality of our proposed framework in diverse decision-making environments.
>
> |FrankaKitchen|Partial|Mixed|
> |:-:|:-:|:-:|
> |Diffuser|$56.2 \pm 5.4$|$50.0 \pm 8.8$|
> |HD|$73.3 \pm 1.4$|$71.7 \pm 2.7$|
> |SIHD|$76.8 \pm 1.7$|$74.0 \pm 2.3$|
>
> |NeoRL|FinRL-L-99|FinRL-L-999|FinRL-M-99|FinRL-M-999|
> |:-:|:-:|:-:|:-:|:-:|
> |MB-PPO|$328$|$656$|$1213$|$698$|
> |HDMI|$415$|$733$|$1007$|$754$|
> |SIHD|$457$|$760$|$1294$|$772$|
>
> $\bullet$ Q1: In Sections 5.3 (Ablation Study) and Appendix C.1 (Sensitivity Analysis), we have already evaluated simplified two-layer diffusion variants of our model—SIHD-DH (with a fixed time horizon) and SIHD-2 (with structure entropy but limited to two levels). The results, summarized in the table below, show that both variants perform noticeably worse than the full SIHD model. This confirms the quantitative advantage of a deeper hierarchical diffusion structure over simpler two-layer architectures.
>
> |AntMaze|U-Maze|Medium|Large|
> |:-:|:-:|:-:|:-:|
> |SIHD-DH|$93.1 \pm 3.8$|$88.6 \pm 5.4$|$85.2 \pm 4.7$|
> |SIHD-2|$93.5 \pm 3.0$|$89.8 \pm 4.7$|$86.1 \pm 4.5$|
> |SIHD|$96.5 \pm 2.8$|$92.2 \pm 5.0$|$89.4 \pm 4.2$|
>
> Additionally, in the Qualitative Comparison section, we extended Figure 7 to present visualizations of hierarchical subgoal generation on the Maze2D navigation task. Compared to HDMI, SIHD produces subgoals with more flexible and longer-range temporal transitions: low‑level diffuser generates fine-grained waypoints that navigate local obstacles, mid‑level diffuser proposes intermediate subgoals that bridge across small clusters of states—e.g., navigating between adjoining corridors, and high‑level diffuser identifies long‑range landmarks or critical ``turning points" across the entire maze, allowing for flexible, multi‑scale planning. This illustrates how the hierarchical design enables more globally coherent planning behavior.
>
> Due to the limitations of the rebuttal format, we are unable to present the updated Figure 7 here, but we will include it in the camera-ready version after conference acceptance for improved clarity and illustration.
>
> $\bullet$ Q2: The use of an encoding tree for hierarchical partitioning is grounded in the principle of structural information theory, which enables hierarchical decomposition of graph nodes without relying on any task-specific prior knowledge. This approach has been validated across various domains [4-6] for its effectiveness in capturing meaningful multi-level structure.
>
> Importantly, the encoding tree supports a well-defined metric—structural entropy—which quantitatively evaluates the quality of the hierarchical partitioning. Unlike standard Shannon entropy, structural entropy promotes not only diversity among nodes but also preservation of the hierarchical community structure. This advantage is formally established in Theorem 4.2 of our paper.
>
> By leveraging the encoding tree and structural entropy, our framework enables a flexible and general multi-level trajectory diffusion process. Furthermore, the integration of structural entropy as a regularizer promotes diverse exploration under the hierarchical structure, which enhances both model performance and generalization.
>
> References:
>
> [1] Hierarchical diffusion for offline decision making. ICML 2023.
>
> [2] Simple hierarchical planning with diffusion. ICLR 2024.
>
> [3] NeoRL: A near real-world benchmark for offline reinforcement learning. NeurIPS 2022.
>
> [4] Robustness Evaluation of Graph-based News Detection Using Network Structural Information. SIGKDD 2025.
>
> [5] Effective Exploration Based on the Structural Information Principles. NeurIPS 2024.
>
> [6] Incremental measurement of structural entropy for dynamic graphs. AIJ 2024.

---

> > ### Comment · Reviewer_42zk · 2025-08-04
> >
> > Thank the authors for the response. I decide to maintain my positive ratings.

---

> > > ### Author Response · Authors · 2025-08-05
> > >
> > > Dear Reviewer,
> > >
> > > Thank you very much for taking the time to read our rebuttal and for confirming that you will keep your positive ratings. We appreciate your constructive feedback throughout the review process.
> > >
> > > Best regards,
> > >
> > > Submission7053 Authors

---

### Note · Authors · 2025-08-13

Dear NeurIPS 2025 Program Chairs, Senior Area Chairs, Area Chairs, and Reviewers,

We would like to extend our heartfelt thanks to everyone who has contributed to the review process of our submission.

Throughout the rebuttal phase we provided detailed, point-by-point responses to every concern raised by the reviewers, and we incorporated corresponding revisions into our manuscript.
These revisions focused on (i) clarifying model details, (ii) expanding comparative evaluations, and (iii) refining formulas and figures for better clarity.
With the assistance of our Area Chair, the subsequent discussion phase enabled an efficient and fruitful exchange.
By the close of discussion, we had successfully addressed virtually all of the reviewers’ questions, and every reviewer ultimately expressed a positive stance toward our work.

Reviewer uo2i, while sharing this positive view, raised a final concern regarding real-world applicability.
We promptly responded with a detailed explanation and expressed our willingness to continue the conversation; unfortunately, we did not receive a follow-up reply before the discussion deadline.
The additional clarifications on this point were provided both in our message to Reviewer uo2i and in our note to the AC.

Finally, we reiterate our commitment: every adjustment and addition described in our rebuttal will appear in the camera-ready version.
We sincerely hope that our revised manuscript meets your expectations and will be included in the NeurIPS 2025 proceedings.
Thank you once again for your guidance and support.

With gratitude,

Submission7053 Authors

---

### Decision · Program_Chairs · 2025-09-17

**Decision:**

Accept (poster)

**Comment:**

This work proposes SIHD, a hierarchical diffusion method for effective and stable offline policy learning in long-horizon environments with sparse rewards. It extracts structural information from a similarity-guided topological graph to build tree-structured communities. And then learns different diffusion processes (can be parameter-shared or a shared diffusion model) for each tree layer. The structural information gain of each state community is used as conditional guidance information. In this work, theoretical guarantees are provided for the decomposition of the conditional generation problem into a hierarchical diffusion process with dynamic temporal scales. Moreover, it uses a structural entropy regularizer to promote exploration of underrepresented states. Experimental results on D4RL, FrankaKitchen, NeoRL demonstrate that the proposed method is promising.

The strengths of this work are listed as follows.

1. Long-horizon tasks with sparse rewards in offline RL are challenging and well motivated.
2. Compared to typical two-layer diffusion processes, a tree-structure diffusion method is novel.
3. The experimental results justify the effectiveness of SIHD.

In the final version, please include the added experimental results and improve the writing of this work (especially the notations).